# CONTRASTIVE AUDIO-VISUAL MASKED AUTOENCODER

**Yuan Gong**[1] (yuangong@mit.edu)**, Andrew Rouditchenko**[1]
**Alexander H. Liu**[1]**, David Harwath**[2]**, Leonid Karlinsky**[3,4]**, Hilde Kuehne**[4,5]**, James Glass**[1]
[1]MIT CSAIL; [2]UT Austin; [3]IBM Research AI; [4]MIT-IBM Watson AI Lab; [5]Goethe University Frankfurt

## ABSTRACT

In this paper, we first extend the recent Masked Auto-Encoder (MAE) model from a single modality to audio-visual multi-modalities. Subsequently, we propose the Contrastive Audio-Visual Masked Auto-Encoder (CAV-MAE) by combining contrastive learning and masked data modeling, two major self-supervised learning frameworks, to learn a joint *and* coordinated audio-visual representation.

Our experiments show that the contrastive audio-visual correspondence learning objective not only enables the model to perform audio-visual retrieval tasks, but also helps the model learn a better joint representation. As a result, our fully self-supervised pretrained CAV-MAE achieves a new SOTA accuracy of 65.9% on VGGSound, and is comparable with the previous best supervised pretrained model on AudioSet in the audio-visual event classification task. Code and pretrained models are at https://github.com/yuangongnd/cav-mae.

## 1 INTRODUCTION

Acoustic and visual modalities have different properties, yet humans are able to seamlessly connect and integrate them to perceive the world. Developing learning algorithms to replicate these abilities, especially for multi-modal audio-visual fusion and retrieval is of great interest. Since manually annotating audio and video is expensive and difficult to scale, how to utilize web-scale *unlabeled* video data in a *self-supervised* manner has become a core research question.

One major line of audio-visual self-supervised learning research is leveraging the natural audio-visual correspondences found in videos. Among numerous ways to use such correspondences, *Contrastive Audio-Visual Learning* has shown to be a simple yet effective approach (Arandjelovic & Zisserman, 2018; Morgado et al., 2021b; Rouditchenko et al., 2021). It learns *coordinated* [1] representations that are closer for paired audio and visual samples than for mismatched samples. Such *coordinated* representations are particularly useful for tasks such as cross-modal retrieval.

Another vetted commonly used self-supervised learning framework is *Masked Data Modeling (MDM)*, which learns a meaningful representation with the pretext task of recovering the original inputs or features from the corrupted ones (Devlin et al., 2019). Particularly, based on the Audio Spectrogram Transformer (Gong et al., 2021a) and Vision Transformer (Dosovitskiy et al., 2020) backbones, the single-modal *Masked Auto-Encoder (MAE)* (He et al., 2022) achieved state-of-the-art (SOTA) performance on images and audio tasks (Huang et al., 2022a) individually. Inspired by these advances, we propose to extend the single-modal MAE to *Audio-Visual Masked Auto-Encoder* (AV-MAE), aiming to learn a *joint* representation that fuses the unimodal signals.

Although these two major self-supervised frameworks have been widely used individually, to the best of our knowledge, they have never been combined in audio-visual learning. In fact, we find they are complementary: Contrastive audio-visual learning explicitly leverages the very useful audio-visual pair information, but it could discard modality-unique information that is useful in downstream tasks; The reconstruction task of AV-MAE forces its representation to encode the majority of the input information in the fusion, but it lacks an explicit audio-visual correspondence objective.

---

[1]Multi-modal representations can be divided into two categories: *joint* representations that combine the unimodal signals into the same representation space, and *coordinated* representations that process unimodal signals separately, but enforce certain similarity constraints on them. (Baltrušaitis et al., 2018)

This motivates us to design the *Contrastive Audio-Visual Masked Autoencoder (CAV-MAE)* that integrates contrastive learning and masked data modeling which learns a joint *and* coordinated audio-visual representation with a single model. Our experiments support our design: on audio-visual event classification, CAV-MAE significantly outperforms baseline models trained with only contrastive or masked data modeling objectives, demonstrating that the two objectives are complementary in learning a strong joint audio-visual representation. As a result, CAV-MAE achieves a new SOTA accuracy of 65.9% on VGGSound, and is comparable with the previous best supervised pretrained model on AudioSet. Moreover, when it comes to audio-visual retrieval, CAV-MAE also performs equally well or even better than models trained with only the contrastive objective, which demonstrates that CAV-MAE can learn both a joint *and* coordinated representation well. Finally, CAV-MAE multi-modal pretraining improves single-modal performance, consequently, CAV-MAE achieves a new SOTA for audio-based event classification on AudioSet-20K and VGGSound.

In summary, our contributions are: (1) We extend the single-modal MAE to multi-modal AV-MAE, which fuses audio-visual inputs for self-supervised learning through cross-modal masked data modeling; (2) More importantly, we investigate how to best combine contrastive audio-visual learning with masked data modeling and propose CAV-MAE; (3) We demonstrate that contrastive and masked data modeling objectives are complementary. As a result, CAV-MAE matches or outperforms SOTA models on audio-visual classification.

## 2 CONSTRASTIVE AUDIO-VISUAL MASKED AUTOENCODER

### 2.1 PRELIMINARIES

#### 2.1.1 AUDIO AND IMAGE PRE-PROCESSING AND TOKENIZATION

As depicted in Figure 1 (A), we follow pre-processing and tokenization in AST (Gong et al., 2021a) and ViT (Dosovitskiy et al., 2020) for audio and image inputs, respectively. Specifically, we use 10-second videos (with parallel audios) in AudioSet (Gemmeke et al., 2017) and VGGSound (Chen et al., 2020) to pretrain and fine-tune the model. For audio, each 10-second audio waveform is first converted to a sequence of 128-dimensional log Mel filterbank (fbank) features computed with a 25ms Hanning window every 10ms. This results in a $1024(\text{time}) \times 128(\text{frequency})$ spectrogram. We then split the spectrogram into 512 $16 \times 16$ square patches $\mathbf{a} = [a^1, ..., a^{512}]$ as the input of the model. Processing video with Transformer models is expensive and typically requires industrial-level computation resources. To lower the computational overhead and fit our resources, we use a *frame aggregation* strategy. Specifically, we uniformly sample 10 RGB frames from each 10-second video (i.e., 1 FPS). During training, we randomly select one RGB frame as the input; during inference, we average the model prediction of each RGB frame as the video prediction. Compare with concatenating multiple RGB frames as the input of the Transformer that has a quadratic complexity (e.g., in Nagrani et al. (2021)), frame aggregation is much more efficient with a linear complexity in time at a cost of not considering inter-frame correlation. For each RGB frame, we resize and center crop it to $224 \times 224$, and then split it into 196 $16 \times 16$ square patches $\mathbf{v} = [v^1, ..., v^{196}]$.

#### 2.1.2 THE TRANSFORMER ARCHITECTURE

Throughout this paper, we use the standard Transformer (Vaswani et al., 2017) as our main model component. Each Transformer layer consists of multi-headed self-attention (MSA), layer normalization (LN), and multilayer perceptron (MLP) blocks with residual connections. Specifically, we denote a Transformer layer $\mathbf{y} = \text{Transformer}(\mathbf{x}; \text{MSA}, \text{LN1}, \text{LN2}, \text{MLP})$ as:

$$\mathbf{x}' = \text{MSA}(\text{LN}_1(\mathbf{x})) + \mathbf{x}; \quad \mathbf{y} = \text{MLP}(\text{LN}_2(\mathbf{x}')) + \mathbf{x}' \tag{1}$$

where MSA computes dot-product attention of each element of $\mathbf{x}$ and thus has a quadratic complexity w.r.t. to the size of $\mathbf{x}$. Please refer to Vaswani et al. (2017) for further details on Transformers.

#### 2.1.3 CONSTRASTIVE AUDIO-VISUAL LEARNING (CAV)

The natural pairing of audio and visual information in videos is a useful signal for learning audio-visual representations through self-supervision. A conventional CAV model is shown in Figure 1.B (top), for a mini-batch of $N$ audio-visual pair samples, we first pre-process and tokenize the audios

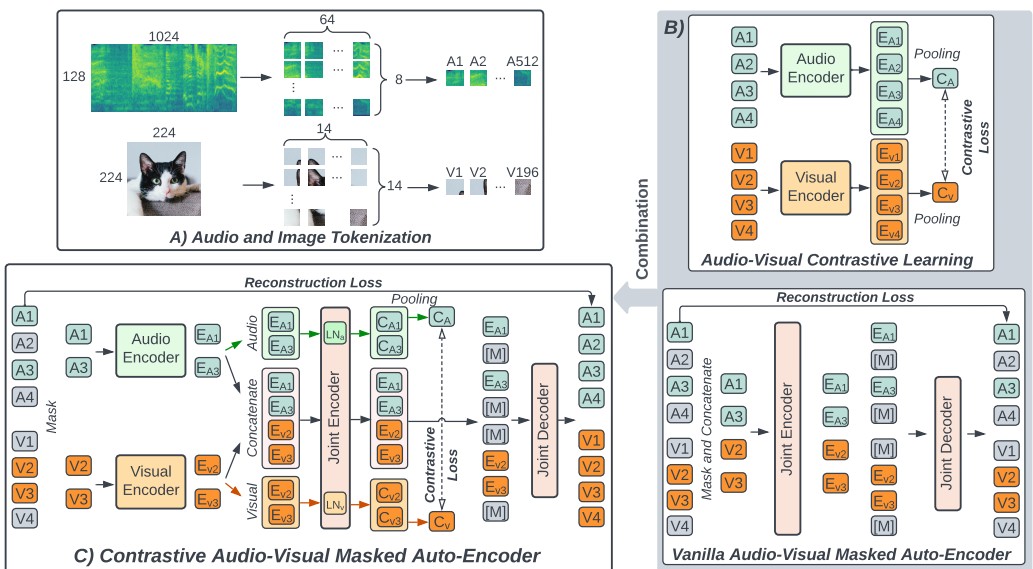

Figure 1: An illustration of our method. A) We tokenize audio spectrograms and RGB images into 16×16 square patches and use them as the input to all models. B) Conventional contrastive audio-visual learning model (top) and vanilla audio-visual masked auto-encoder (bottom, also novel and first introduced in this paper). C) Our proposed contrastive audio-visual masked auto-encoder (CAV-MAE) model. CAV-MAE integrates two major self-supervised frameworks: contrastive audio-visual learning and cross-modal masked data modeling, which learns a *joint and coordinate* representations and performs well on both multi-modal joint classification tasks and cross-modal retrieval tasks.

and images and get a sequence of audio and visual tokens $\{\mathbf{a}_i, \mathbf{v}_i\}$ for each sample $i$. We then input $\mathbf{a}_i$ and $\mathbf{v}_i$ to independent audio and visual Transformer encoders $\mathrm{E}_a(\cdot)$ and $\mathrm{E}_v(\cdot)$, respectively, and get the mean pooled audio and visual representation $c_i^a$ and $c_i^v$, i.e., $c_i^a = \mathrm{MeanPool}(\mathrm{E}_a(\mathrm{Proj}_a(\mathbf{a}_i))$ and $c_i^v = \mathrm{MeanPool}(\mathrm{E}_v(\mathrm{Proj}_v(\mathbf{v}_i))$, where $\mathrm{Proj}_a$ and $\mathrm{Proj}_v$ are linear projections that maps each audio and visual token to $\mathbb{R}^{768}$. We then apply a contrastive loss (Equation 7) on $c_i^a$ and $c_i^v$.

### 2.1.4 SINGLE MODALITY MASKED AUTOENCODER (MAE)

Another line of major self-supervised frameworks is masked data modeling (MDM). Among numerous variants of MDM (e.g., Bao et al. (2021); Wei et al. (2022)), the masked auto-encoder (MAE) is a simple yet effective approach. For an input sample $\mathbf{x}$ that can be tokenized as $\mathbf{x} = [x^1, x^2, ..., x^n]$, MAE *masks* a portion of the input $\mathbf{x}_{\mathrm{mask}}$ and only inputs the unmasked tokens $\mathbf{x} \setminus \mathbf{x}_{\mathrm{mask}}$ to a Transformer based encoder-decoder model. The model is asked to *reconstruct* the masked tokens with the goal of minimizing the mean square error (MSE) loss. During this process, the model learns a meaningful representation of the input data. The advantages of MAE are multifold. First, MAE directly uses the original input as the prediction target, which greatly simplifies the training pipeline. Second, MAE only inputs *unmaksed* tokens to the encoder, and combined with a high masking ratio, MAE noticeably lowers the computational overhead. Third, MAE demonstrated strong performance in single-modal tasks for both audio and visual modalities. Due to the space limitation, please refer to He et al. (2022); Huang et al. (2022a) for single-modal MAEs.

### 2.2 VANILLA AUDIO-VISUAL MASKED AUTOENCODER (AV-MAE)

While MAE has been applied to both audio and visual modality individually, it has never been applied to audio-visual multi-modality learning. As the first contribution of this work, we extend MAE from a single modality to audio-visual multi-modality and build a "vanilla" audio-visual autoencoder (AV-MAE). As shown in Figure 1.B (bottom), for a pair of audio and image inputs, we first tokenize them to $\mathbf{a} = [a^1, ..., a^{512}]$ and $\mathbf{v} = [v^1, ..., v^{196}]$ and project them to $\mathbb{R}^{768}$ with two modal-specific linear projection layer as well as add a modality type embedding $\mathbf{E_a}$ and $\mathbf{E_v}$ and modality

specific 2-D sinusoidal positional embedding $\mathbf{E_a^P}$ and $\mathbf{E_v^P}$, i.e., $\mathbf{a}' = \mathrm{Proj_a}(\mathbf{a}) + \mathbf{E_a} + \mathbf{E_a^P}$ and $\mathbf{v}' = \mathrm{Proj_v}(\mathbf{v}) + \mathbf{E_v} + \mathbf{E_v^P}$. We concatenate $\mathbf{a}'$ and $\mathbf{v}'$ and construct a joint embedding $\mathbf{x} = [\mathbf{a}', \mathbf{v}']$. We then mask a portion (75%) of $\mathbf{x}$ and only input unmasked tokens $\mathbf{x_{unmask}} = \mathbf{x} \setminus \mathbf{x_{mask}}$ to an audio-visual joint encoder $\mathrm{E_j}(\cdot)$ and get the output $\mathbf{x'_{unmask}}$. After that, we pad $\mathbf{x'_{unmask}}$ with trainable masked tokens at their original position as $\mathbf{x}'$. Again, we also add modality type embedding $\mathbf{E'_a}$ and $\mathbf{E'_v}$ and modality-specific 2-D sinusoidal positional embedding $\mathbf{E_a^{P'}}$ and $\mathbf{E_v^{P'}}$ before feeding $\mathbf{x}'$ to a joint audio-visual decoder $\mathrm{D_j}(\cdot)$ to reconstruct the input, i.e., $\hat{\mathbf{a}}, \hat{\mathbf{v}} = \mathrm{D_j}(\mathbf{x}' + [\mathbf{E'_a}, \mathbf{E'_v}] + [\mathbf{E_a^{P'}}, \mathbf{E_v^{P'}}])$ Finally, we minimize the mean square error (MSE) between $\hat{\mathbf{a}}, \hat{\mathbf{v}}$ and normalized $\mathbf{a}, \mathbf{v}$.

Compared with single-modal MAEs, the AV-MAE features a *cross-modal masked data modeling* objective that allows the model to reconstruct one modality based on the information of another modality, which may help the model learn audio-visual correlation. However, without an explicit objective of encouraging paired audio-visual correspondence, vanilla AV-MAE actually does not effectively leverage the audio-visual pairing information (discussed in Appendix J). Also, using a joint encoder for two modalities allows cross-modal attention, but it also means the two very different modalities are processed with the same weights, which could lead to a sub-optimal solution.

## 2.3 CONSTRASTIVE AUDIO-VISUAL MASKED AUTOENCODER (CAV-MAE)

As discussed in Section 2.1.3 and 2.2, contrastive audio-visual learning and AV-MAE each has its advantages and disadvantages. Can we integrate the complementary advantages of CAV and AV-MAE? With this goal, we design the Contrastive Audio-Visual Masked Autoencoder (CAV-MAE) (shown in Figure 1.C). For a mini-batch of $N$ audio-visual pair samples, we first pre-process and tokenize the audios and images and get a sequence of audio and visual tokens $\{\mathbf{a}_i, \mathbf{v}_i\}$ for each sample $i$ and project them to $\mathbb{R}^{768}$ with two modal-specific linear projection layer. We also add a modality type embedding $\mathbf{E_a}$ and $\mathbf{E_v}$ and modality-specific 2-D sinusoidal positional embedding $\mathbf{E_a^P}$ and $\mathbf{E_v^P}$. After that, we uniformly mask 75% of tokens of each modality, i.e.,

$$\mathbf{a}_i^{\mathrm{unmask}} = \mathrm{Mask_{0.75}}(\mathrm{Proj_a}(\mathbf{a}_i) + \mathbf{E_a} + \mathbf{E_a^P}) \tag{2}$$

$$\mathbf{v}_i^{\mathrm{unmask}} = \mathrm{Mask_{0.75}}(\mathrm{Proj_v}(\mathbf{v}_i) + \mathbf{E_v} + \mathbf{E_v^P}) \tag{3}$$

We then input $\mathbf{a}_i^{\mathrm{unmask}}$ and $\mathbf{v}_i^{\mathrm{unmask}}$ to independent audio and visual Transformer encoders $\mathrm{E_a}(\cdot)$ and $\mathrm{E_v}(\cdot)$ and get $\mathbf{a}'_i$ and $\mathbf{v}'_i$, respectively. After that, we apply *multi-stream* forward passes to input $\mathbf{a}'_i, \mathbf{v}'_i$ to a joint audio-visual encoder $\mathrm{E_j}(\cdot; \mathrm{MSA, LN1, LN2, MLP})$. Specifically, we input audio tokens $\mathbf{a}'_i$, video tokens $\mathbf{v}'_i$, and concatenated audio-visual tokens $[\mathbf{a}'_i, \mathbf{v}'_i]$ in three independent forward passes to $\mathrm{E_j}$. For each stream, we use different layer normalization layers $\mathrm{LN1_{\{a,v,av\}}}$ and $\mathrm{LN2_{\{a,v,av\}}}$, all other weights (i.e., weights of the MSA and MLP) of $\mathrm{E_j}$ are shared for all three streams. Formally,

$$c_i^a = \mathrm{MeanPool}(\mathrm{E_j}(\mathrm{E_a}(\mathbf{a}_i^{\mathrm{unmask}})); \mathrm{LN1_a, LN2_a})) \tag{4}$$

$$c_i^v = \mathrm{MeanPool}(\mathrm{E_j}(\mathrm{E_v}(\mathbf{v}_i^{\mathrm{unmask}})); \mathrm{LN1_v, LN2_v})) \tag{5}$$

$$\mathbf{x_i} = \mathrm{E_j}([\mathrm{E_a}(\mathbf{a}_i^{\mathrm{unmask}}), \mathrm{E_v}(\mathbf{v}_i^{\mathrm{unmask}})]; \mathrm{LN1_{av}, LN2_{av}}) \tag{6}$$

We use the output of the audio and visual single modality stream $c_i^a$ and $c_i^v$ for contrastive learning and the output of the audio-visual multi-modal stream $\mathbf{x_i}$ for the reconstruction task.

For contrastive audio-visual learning, we use the contrastive loss $\mathcal{L}_c$:

$$\mathcal{L}_c = -\frac{1}{N} \sum_{i=1}^{N} \log \left[ \frac{\exp(s_{i,i}/\tau)}{\sum_{k \neq i} \exp(s_{i,k}/\tau) + \exp(s_{i,i}/\tau)} \right] \tag{7}$$

where $s_{i,j} = \|c_i^v\|^T \|c_j^a\|$ and $\tau$ is the temperature.

For the reconstruction task, we pad $\mathbf{x_i}$ with trainable masked tokens at their original position as $\mathbf{x'_i}$. We also add modality type embedding $\mathbf{E'_a}$ and $\mathbf{E'_v}$ and modality-specific 2-D sinusoidal positional embedding $\mathbf{E_a^{P'}}$ and $\mathbf{E_v^{P'}}$ before feeding $\mathbf{x'_i}$ to a joint audio-visual decoder $\mathrm{D_j}(\cdot)$ to reconstruct the input audio and image. $\mathrm{D_j}(\cdot)$ processes audio and visual tokens with a same set of weights except the last modal-specific projection layer, it outputs $\hat{\mathbf{a}}_i$ and $\hat{\mathbf{v}}_i$. We then apply a mean square error reconstruction loss $\mathcal{L}_r$:

$$\hat{\mathbf{a}}_i, \hat{\mathbf{v}}_i = \mathrm{D_j}(\mathbf{x}' + [\mathbf{E'_a}, \mathbf{E'_v}] + [\mathbf{E_a^{P'}}, \mathbf{E_v^{P'}}]) \tag{8}$$

$$\mathcal{L}_r = \frac{1}{N} \sum_{i=1}^{N} \left[ \frac{\sum (\hat{\mathbf{a}}_i^{\text{mask}} - \text{norm}(\mathbf{a}_i^{\text{mask}}))^2}{|\mathbf{a}_i^{\text{mask}}|} + \frac{\sum (\hat{\mathbf{v}}_i^{\text{mask}} - \text{norm}(\mathbf{v}_i^{\text{mask}}))^2}{|\mathbf{v}_i^{\text{mask}}|} \right] \qquad (9)$$

where $N$ is the mini-batch size; $\mathbf{a}^{\text{mask}}$, $\mathbf{v}^{\text{mask}}$, $\hat{\mathbf{a}}^{\text{mask}}$, $\hat{\mathbf{v}}^{\text{mask}}$ denote the original and predicted masked patches (we only calculate the loss based on the masked portion of the input); $|\mathbf{a}^{\text{mask}}|$ and $|\mathbf{v}^{\text{mask}}|$ denote the number of masked audio and visual patches, respectively.

Finally, we sum up the contrastive loss $\mathcal{L}_c$ (multiplied by a weight $\lambda_c$) and the reconstruction loss $\mathcal{L}_r$ as the loss for CAV-MAE, i.e., $\mathcal{L}_{\text{CAV-MAE}} = \mathcal{L}_r + \lambda_c \cdot \mathcal{L}_c$.

After pretraining, we abandon the decoder and only keep the encoders of the model for downstream tasks. We can use the sum of the single-modality stream output and the multi-modal modality stream output, or just the multi-modal stream output for finetuning. They perform similarly in our experiments.

**Discussion:** we next discuss the motivation of some key designs of CAV-MAE:

*1. Multi-stream forward passes of the joint encoder.* We find it important to restrict the representations used for contrastive audio-visual learning, so that $c^a$ only comes from the audio input and $c^v$ only comes from the visual input, otherwise the contrastive objective will collapse. In the meantime, we hope the encoder fuses the audio and visual information for the reconstruction task and downstream tasks. Therefore, we design the multi-stream forward pass strategy for CAV-MAE.

*2. Modality-specific encoders and* LN *layers.* While there are a few recent attempts (Akbari et al., 2021; Dai et al., 2022) to process audio and visual modalities with a unified network, due to the very different nature of audio and visual modalities, the general conclusion is that modality-specific networks are still optimal in terms of performance. Therefore, we choose to encode audio and visual inputs with modality-specific encoders before the joint encoder. For the same reason, we also use different normalization statistics for each stream of the joint encoder. Efficiency-wise, having two modality-specific encoders increases the model size, but lowers the computation as the Transformer has a quadratic complexity w.r.t. the input sequence length.

*3. Masked contrastive audio-visual learning.* Unlike single-modality contrastive learning, conventional contrastive audio-visual learning does not typically apply augmentation or masking. In this work, we propose to use *masked contrastive audio-visual learning*, i.e., we randomly mask a portion of the input before conducting contrastive learning. This design not only allows us to combine CAV with AV-MAE, but also helps to avoid overfitting. In practice, when the masking ratio is 75% and the effective contrastive batch size is 27 (108 on 4 GPUs), the audio-visual matching accuracy during pretraining on the evaluation set is about 72%, which shows the task is neither trivial nor impossible. We discuss the impact of masking on contrastive learning in detail in Appendix F.

### 2.3.1 Implementation Details

By default, all encoder Transformer layers are 768-dimensional and have 12 attention heads. The joint encoder of the Vanilla AV-MAE is a 12-layer Transformer; The audio and visual encoders of CAV-MAE are 11-layer Transformers (each is 768- dimensional) and the joint encoder is a single-layer Transformer. I.e., we control the total number of encoder layers of all models as 12, but CAV and CAV-MAE are larger models due to the modality-specific encoders. The decoder of AV-MAE and CAV-MAE are 8-layer Transformers with an embedding dimension of 512 and 16 attention heads. These settings are identical to the original vision MAE He et al. (2022). We fix the contrastive loss temperature $\tau = 0.05$. For CAV-MAE, we use $\lambda_c = 0.01$. Note the relatively small $\lambda_c$ is due to the scale of the gradient of $\mathcal{L}_c$ being larger than $\mathcal{L}_r$, it does not mean the contrastive objective is unimportant. The encoder and decoder of the default CAV-MAE model have about 164M and 27M parameters, respectively.

Following the common practice of audio-visual learning, we initialize the weights of all models with ImageNet pretrained weights. Specifically, we use the weights of the original vision MAE He et al. (2022). Nevertheless, unlike previous work that uses supervised pretrained weights (e.g., Fayek & Kumar (2021) and Nagrani et al. (2021)), we only use the self-supervised pretrained weights (i.e., without finetuning), which does not lead to the best performance but makes our whole training pipeline self-supervised. The impact of initialization strategy is discussed in detail in Appendix E.

## 3 SELF-SUPERVISED MODEL PRETRAINING

We pretrain and compare the performance of the following models:

1. **Audio-MAE/Visual-MAE:** Single-modal masked auto-encoder models. The model architecture is the same with `Vanilla AV-MAE` but they are only pretrained with data of a single modality.
2. **CAV:** The contrastive audio-visual learning model that has no reconstruction objective. For a fair comparison, we implement `CAV` using the same encoder architecture (modal-specific encoders + joint encoder) with `CAV-MAE` but remove the reconstruction objective $\mathcal{L}_r$.
3. **Vanilla AV-MAE:** The vanilla audio-visual masked auto-encoder with a joint encoder and no contrastive objective as described in Section 2.2.
4. **AV-MAE:** The audio-visual masked auto-encoder with two modal-specific encoders and a joint encoder. It has the same architecture with `CAV-MAE`, but $\lambda_c$ is set to 0 (no contrastive loss). We use this model to disentangle the impact of modal-specific encoders (when compared with `Vanilla AV-MAE`) and contrastive objective (when compared with `CAV-MAE`).
5. **CAV-MAE:** Our proposed contrastive masked auto-encoder as described in Section 2.3.
6. **CAV-MAE$^{scale+}$:** The same model with `CAV-MAE`, but trained with a larger batch size=108 (effective contrastive batch size=27) and more epochs=25. We train this model on our best GPUs.

For a fair comparison, all models (except `CAV-MAE`$^{scale+}$) are pretrained with the same pipeline with a batch size of 48 for 12 epochs on the full AudioSet-2M. During pretraining, we intentionally do not use class balanced sampling as that implicitly leverages the label information. Our pretraining process (including the ImageNet pretrained weight initialization) is fully self-supervised. Please refer to Appendix B for all pretraining details.

## 4 AUDIO-VISUAL EVENT CLASSIFICATION

We evaluate the representation quality on the audio-visual event classification task, a major audio-visual learning benchmark. Specifically, we fine-tune the pretrained models on three datasets: 1) AudioSet-20K (20K samples, same domain as the pretraining data); 2) AudioSet-2M (2 million samples, same with pretraining data); and 3) VGGSound (200K samples, different domain than the pretraining data), covering various downstream data volume and domain situations.

In the fine-tuning stage, we only keep the encoder of the pretrained models and connect it to a randomly initialized linear classification head. To avoid overriding too much of the knowledge learned in pretraining, we use a smaller learning rate for the pretrained weights and a 10×-100× larger learning rate for the new classification head. We use the standard training pipeline used in prior audio-based and audio-visual event classification work Gong et al. (2021a;b); Nagrani et al. (2021) with mixup Zhang et al. (2018), balanced sampling, label smoothing, label enhancement (only for AudioSet-20K) and random time shifts. We fine-tune the model using audio-only data (A), video-only data (V), and audio-visual data (AV) to evaluate the single-modal and multi-modal representation quality. We show the results in Table 1. Key findings are as follows:

**1. Contrastive learning and masked data modeling are complementary.** While both `AV-MAE` (only with masked data modeling objective) and `CAV` (only with contrastive objective) perform better than ensembling two single-modal MAEs, the proposed `CAV-MAE` that combines the two objectives significantly boosts the performance (e.g., 2.0 and 3.1 mAP boost from `CAV` and `AV-MAE` on AudioSet-20K, respectively). Note `CAV-MAE`, `AV-MAE`, and `CAV` have the same architecture during fine-tuning, the only difference is the objective in the pretraining stage. This demonstrates that the two major self-supervised learning frameworks are complementary in the context of audio-visual learning and CAV-MAE is an effective way to combine their advantages.

**2. CAV-MAE multi-modal pretraining improves single-modal performance.** We find the `CAV-MAE` model pretrained with paired audio-visual data, when fine-tuned with only a single modality, performs noticeably better than `Audio-MAE` and `Visual-MAE` on single-modal classification tasks (e.g., 34.2→37.7 mAP for audio, 15.7→19.8 mAP for visual on AudioSet-20K). Note for single-modal fine-tuning, `CAV-MAE` only keeps one branch and has the same architecture with `Audio-MAE` and `Visual-MAE`, so the performance improvement can only come from the use of multi-modal data during pretraining. We hypothesize this is due to the two modalities serving as soft labels for each other, providing richer information than the binary human-annotated labels.

Table 1: Comparing audio-visual classification performance on AudioSet and VGGSound. IN SL=ImageNet supervised learning; SSL=self-supervised learning; †Industry-level computation. *Nonstandard data split; ensEnsemble of single-modal models. We bold the best methods without supervised pretraining, and underline the overall best methods.

| | Pretrain | AudioSet-20K (mAP) | | | AudioSet-2M (mAP) | | | VGGSound (Acc) | | |
|---|---|---|---|---|---|---|---|---|---|---|
| | | A | V | A-V | A | V | A-V | A | V | A-V |
| ***Existing Audio-Based Models*** | | | | | | | | | | |
| PANNs (Kong et al., 2020) | - | 27.8 | - | - | 43.9 | - | - | - | - | - |
| AST (Gong et al., 2021a) | IN SL | 34.7 | - | - | 45.9 | - | - | - | - | - |
| HTS-AT Chen et al. (2022) | IN SL | - | - | - | 47.1 | - | - | - | - | - |
| PaSST Koutini et al. (2021) | IN SL | - | - | - | 47.1 | - | - | - | - | - |
| SSAST (Gong et al., 2022) | SSL | 31.0 | - | - | - | - | - | - | - | - |
| MAE-AST (Baade et al., 2022) | SSL | 30.6 | - | - | - | - | - | - | - | - |
| Audio-MAE†(vanilla) (Huang et al., 2022a) | SSL | 36.6 | - | - | 46.8 | - | - | - | - | - |
| Audio-MAE† (Huang et al., 2022a) | SSL | 37.1 | - | - | 47.3 | - | - | - | - | - |
| Chen et al. (2020) | - | - | - | - | - | - | - | 48.8 | - | - |
| AudioSlowFast (Kazakos et al., 2021) | - | - | - | - | - | - | - | 50.1 | - | - |
| ***Existing Audio-Visual Models*** | | | | | | | | | | |
| GBlend†* (Wang et al., 2020) | - | 29.1 | **22.1** | 37.8 | 32.4 | 18.8 | 41.8 | - | - | - |
| Perceiver† (Jaegle et al., 2021) | - | - | - | - | 38.4 | 25.8 | 44.2 | - | - | - |
| Attn AV (Fayek & Kumar, 2021) | IN SL | - | - | - | 38.4 | 25.7 | 46.2 | - | - | - |
| MBT†* (Nagrani et al., 2021) | IN SL | 31.3 | 27.7 | 43.9 | 44.3 | 32.3 | 52.1 | 52.3 | 51.2 | 64.1 |
| ***Our Single-Modal MAE*** | | | | | | | | | | |
| Audio-MAE | SSL | 34.2 | - | 36.7ens | 44.9 | - | 46.9ens | 57.7 | - | 63.1ens |
| Visual-MAE | SSL | - | 15.7 | | - | 24.2 | | - | 45.7 | |
| ***Our Contrastive Audio-Visual Learning*** | | | | | | | | | | |
| CAV | SSL | 34.6 | 18.4 | 38.5 | 43.6 | 24.4 | 48.1 | 57.3 | 45.1 | 64.1 |
| ***Our Multi-Modal MAE*** | | | | | | | | | | |
| Vanilla AV-MAE | SSL | 32.7 | 15.8 | 36.5 | 43.7 | 24.0 | 48.3 | 56.4 | 45.4 | 63.4 |
| AV-MAE | SSL | 33.4 | 15.1 | 37.4 | 44.8 | 24.0 | 49.6 | 57.2 | 45.3 | 64.1 |
| CAV-MAE | SSL | 36.8 | 18.7 | 40.5 | 45.8 | 25.6 | 50.5 | 59.2 | 46.6 | 65.4 |
| CAV-MAE$^{Scale+}$ | SSL | **37.7** | 19.8 | **42.0** | 46.6 | **26.2** | **51.2** | **59.5** | **47.0** | **65.5** |

As a result, `CAV-MAE` achieves a new SOTA performance on audio-based event classification on AudioSet-20K (37.7 mAP) and VGGSound (59.5% accuracy), without supervised pretraining and industry-level computational resources.

**3. Fully SSL pretrained CAV-MAE matches or outperforms SOTA models with significantly fewer computational resources.** There are two major setting differences between this work and previous SOTA works. First, our pretraining is completely self-supervised so that our model can leverage web-scale unlabeled videos, while supervised ImageNet pretraining is commonly used in previous audio-visual works, e.g., in MBT (Nagrani et al., 2021). ImageNet labels are strong supervision signals that can directly impact the visual branch performance (see Table 11). As a result, our visual branch is worse than the SOTA models. Second, we pretrain and fine-tune the model with 4 GPUs (which also makes our work easy to reproduce), while most SOTA models are trained with industry-level resources (e.g., 32 TPUs for Perceiver (Jaegle et al., 2021), 64 GPUs for Audio-MAE (Huang et al., 2022a) and MBT), which brings many benefits such as large batch size (particularly useful for contrastive learning), multiple frames input (MBT uses 8 frames as input), and more training epochs (Audio-MAE pretrains the model for 32 epochs).

Even with such setting differences, on the audio-visual event classification task, our `CAV-MAE` performs better than the best existing audio-visual model MBT on VGGSound (even when `CAV-MAE` is only pretrained on VGGSound, see Table 2e) and comparable on AudioSet-20K and AudioSet-2M. On the audio-based event classification task, our `CAV-MAE` performs better than the best existing audio model Audio-MAE on AudioSet-20k and comparable on AudioSet-2M.

Besides, we find modal-specific encoders are helpful as `AV-MAE` outperforms `Vanilla AV-MAE`. `Vanilla AV-MAE` with only a joint encoder does not outperform the ensemble of single-modal `Audio-MAE` and `Visual-MAE`. Scaling up the batch size and training epochs improves the performance as `CAV-MAE`scale+ generally performs better than `CAV-MAE`. The performance margin is smaller on larger fine-tuning datasets. Finally, We also evaluate the models on the audio-visual action recognition task (Appendix C), which leads to consistent conclusions.

Table 2: Ablation studies on audio-visual classification. MM=multi-modal, SM=single-modal.

| (a) **Pretrain** $\lambda_c$ | |
| --- | --- |
| $\lambda_c$ | AS-20K |
| 0.1 | 39.3 |
| **0.01** | **40.5** |
| 0.001 | 38.6 |

| (b) **Pretrain epochs** | |
| --- | --- |
| Epochs | AS-20K |
| 1 | 37.3 |
| 3 | 39.1 |
| 12 | 40.8 |
| **25** | **42.0** |

| (c) **Pretrain batch** | |
| --- | --- |
| Size | AS-20K |
| 48 | 40.5 |
| **108** | **40.8** |

| (d) **Pretrain target** | |
| --- | --- |
| Norm | AS-20K |
| w/o norm | 40.5 |
| **w/norm** | **40.5** |

| (e) **Pretrain dataset** | |
| --- | --- |
| Dataset | VGGSound |
| AS-2M | 65.5 |
| VS | 64.2 |
| **AS-2M+VS** | **65.9** |

| (f) **Finetuning** | |
| --- | --- |
| Strategy | AS-20K |
| **MM** | **42.0** |
| SM | 41.3 |
| MM+SM | 41.7 |

| (g) **SM experiment** | | |
| --- | --- | --- |
| Setting | AS-20K | |
| | A | V |
| Missing Modality | 36.7 | 14.4 |
| **SM Fine-tune** | **37.7** | **19.8** |

| (h) **Inference frame** | | |
| --- | --- | --- |
| Frame | AS-20K | |
| Used | V | A-V |
| Middle | 17.4 | 40.9 |
| **Aggregation** | **19.8** | **42.0** |

| (i) **Linear probe** | |
| --- | --- |
| Model | AS-20K |
| SM Ensemble | 24.2 |
| AV-MAE | 24.0 |
| **CAV-MAE** | **29.8** |

**Ablation Studies:** We conduct a series of ablation studies to show the impact of each design factor. For each study, we use `CAV-MAE``scale+` or `CAV-MAE` as the base model, change one factor at a time, and report the downstream classification performance of the model on AudioSet-20K or VG-GSound. Our findings are as follows: the weight of the contrastive loss $\lambda_c$ has a large impact on the performance, too large or too small $\lambda_c$ leads to a noticeable performance drop (Table 2a); Scaling up the pretraining epochs and batch size consistently leads to a performance improvement (Table 2b and 2c); Normalizing the prediction target only leads to marginal performance improvement (Table 2d); When finetuning on VGGSound, pretraining with the larger out-of-domain AudioSet-2M is better than pretraining with the smaller in-domain VGGSound itself, but pretraining first on AudioSet-2M and then on VGGSound leads to the best result (Table 2e); During fine-tuning, using the output of the multi-modal stream of the encoder leads to better performance than using the concatenated single-modal stream outputs, and summing the output of two streams generally lead to similar result (Table 2f); When only one modality is of interest, it is better to fine-tune the model with single-modal data than fine-tune the model with audio-visual data and do single modality inference. However, the performance gap is small for audio (Table 2g); The frame aggregation strategy boosts the performance without the need to input multiple frames simultaneously to the model (Table 2h); In the linear probe setting, `CAV-MAE` also noticeably outperform the baselines (Table 2i). We also study the impact of model initialization, masking strategy, and frame rate in Appendix E,F,G, respectively.

## 5 AUDIO-VISUAL RETRIEVAL

Table 3: Retrieval results on AudioSet and VGGSound.

| Visual → Audio | AudioSet Eval Subset | | | VGGSound Eval Subset | | |
| --- | --- | --- | --- | --- | --- | --- |
| | R@1 | R@5 | R@10 | R@1 | R@5 | R@10 |
| *Audio-Visual Models with Only MDM Loss* | | | | | | |
| Vanilla AV-MAE | 0.1 | 0.3 | 0.8 | 0.2 | 0.7 | 1.4 |
| AV-MAE | 0.1 | 0.3 | 0.7 | 0.1 | 0.7 | 1.2 |
| *Audio-Visual Models with Only Contrastive Loss* | | | | | | |
| CAV, $\lambda_c = 0.1$ | **17.4** | 36.1 | 47.3 | 14.2 | 35.2 | **46.2** |
| CAV, $\lambda_c = 0.01$ | 14.6 | 32.9 | 42.8 | 10.9 | 28.7 | 39.8 |
| *Constrastive Audio-Visual Masked Auto-Encoders* | | | | | | |
| CAV-MAE, $\lambda_c = 0.1$ | 16.1 | **38.6** | **49.3** | **14.7** | **35.3** | 45.9 |
| CAV-MAE, $\lambda_c = 0.01$ | 12.3 | 31.4 | 41.9 | 12.5 | 28.6 | 39.1 |
| CAV-MAEScale+, $\lambda_c = 0.01$ | 18.8 | 39.5 | 50.1 | 14.8 | 34.2 | 44.0 |

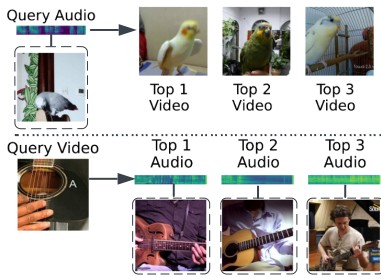

Figure 2: Sample retrieval results.

In the previous section, we show that CAV-MAE learns a good audio-visual *joint* representation that effectively fuses the unimodal signals for the audio-visual event classification task. Next, we study if CAV-MAE also learns a good *coordinated* representation that captures audio-visual correspondences for audio-visual retrieval. Specifically, we uniformly sample a subset of 1,725 and 1,545 audio-visual samples from the AudioSet and VGGSound evaluation set, respectively (about 10%) to make the similarity matrix of a reasonable size. We input audio and image to each model in two independent forward passes and take the mean-pooled encoder outputs as the audio and visual representation, respectively. We then calculate the retrieval recall at rank 1, 5, and 10 (R@1, R@5, R@10)

based on the cosine similarity of the audio and visual representation. All models are self-supervised pretrained but not fine-tuned. We show the quantitative results and samples of visual→audio retrieval in Table 3 and Figure 2, respectively. The results of audio→ visual retrieval, more samples, and additional retrieval experiments on MSR-VTT (Xu et al., 2016) can be found in Appendix D.

We find a contrastive objective is necessary for the audio-visual retrieval task as the performance of both `Vanilla-MAE` and `AV-MAE` are close to random guesses. Nevertheless, the cross-modal masked data modeling objective does not hurt, and in many cases, improves the retrieval performance, e.g., when $\lambda_c = 0.1$, `CAV-MAE` generally performs better than `CAV`. Scaling up the batch size and training epoch also leads to a better retrieval performance. When tested on a dataset different from the pretraining dataset (VGGSound), the retrieval performance is still competitive, indicating the audio-visual correspondence transfers well in addition to the audio and visual representations. These results demonstrate that the contrastive and mask data modeling objectives do not conflict, a single pretrained `CAV-MAE` can be applied to both audio-visual fusion and correspondence tasks.

## 6 RELATED WORK

**Contrastive Audio-Visual Learning** The natural pairing of audio and visual information in videos has been a useful signal for learning audio-visual representations through self-supervision. Existing methods include knowledge distillation (Aytar et al., 2016; Owens et al., 2016), paired sample discrimination (Arandjelovic & Zisserman, 2017; Korbar et al., 2018; Owens & Efros, 2018), and contrastive learning (Morgado et al., 2021b). To improve contrastive learning, some recent methods sought to mine better negative samples (Ma et al., 2020; Morgado et al., 2021a), while others proposed additional data augmentation (Patrick et al., 2021; Wang et al., 2021) or using global and local video views (Zeng et al., 2021; Recasens et al., 2021). Our approach instead combines the contrastive loss with masked data modeling, which not only leads to an improvement in classification performance but also maintains the compelling ability of audio-visual retrieval (Arandjelovic & Zisserman, 2018; Rouditchenko et al., 2021).

**Masked Auto-Encoder**. Masking data modeling has a long history (Vincent et al., 2008) and has been applied on visual and audio domains (Baevski et al., 2020; Hsu et al., 2021; Srivastava et al., 2022) Given the success of MAE in the vision domain (He et al., 2022; Bachmann et al., 2022; Girdhar et al., 2022; Tong et al., 2022; Feichtenhofer et al., 2022), several efforts adapt MAE for audio with relatively minor changes to the overall pipeline (Baade et al., 2022; Niizumi et al., 2022; Chong et al., 2022; Huang et al., 2022a). There are a few recent works investigating multi-modal MAE for the vision & language multi-modal scenarios (Geng et al., 2022; Kwon et al., 2022), which inspired us to design an audio-visual MAE. To the best of our knowledge, our AV-MAE and CAV-MAE are the first audio-visual masked autoencoders. One closely related concurrent work is CMAE (Huang et al., 2022b), which also combines MAE and contrastive loss, but only for single-modal images. Our motivation and implementation are very different from CMAE as we aim to leverage the unique audio-visual pair information and CAV-MAE features a multi-stream joint encoder design. Finally, while we take a modern approach with Transformers, multi-modal autoencoders have been studied more than a decade ago with much simpler models and datasets (Ngiam et al., 2011).

## 7 CONCLUSION

In this paper, we introduce CAV-MAE, a novel audio-visual learning model. The main idea of this paper is simple: masked data modeling and contrastive learning are a pair of complementary frameworks that should be used together for audio-visual self-supervised learning. Effectively combining the two frameworks and avoiding representation collapse requires some careful design such as the multi-stream forward pass strategy, joint-specific encoder architecture, and masked contrastive learning. From the perspective of representation learning, CAV-MAE learns a joint *and* coordinated representation and can be used for both audio-visual joint event classification task as well as the audio-visual retrieval task. As a result, on the audio-visual event classification task, CAV-MAE matches or outperforms SOTA models with fully self-supervised pretraining and noticeably fewer computational resources; on the retrieval task, CAV-MAE is comparable to models trained with only the contrastive objective. Finally, CAV-MAE multi-modal pretraining also learns strong single-modal representations, which leads to a new SOTA performance on audio-based event classification.

**Acknowledgments**: This research is supported by the MIT-IBM Watson AI Lab.

## ETHICS STATEMENT

The data used in this paper are publicly available YouTube videos, we do not use videos that have been removed by the user. The proposed audio-visual model can be applied in a wide range of areas including security-related applications. However, it can also be used for malicious purposes such as surveillance. We are committed to distributing our code and model carefully.

## REPRODUCIBILITY STATEMENT

We document all implementation details in Section 2.3.1 and Appendix B. Code and pretrained models are available at `https://github.com/yuangongnd/cav-mae`.

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

## A  DATASET DETAILS

We use two major audio-visual datasets for our experiments: AudioSet Gemmeke et al. (2017) and VGGSound Chen et al. (2020). AudioSet-2M is a collection of 2M 10-second YouTube video clips labeled with the sounds that the clip contains from a set of 527 labels of audio events, AudioSet-20K is a subset of AudioSet-2M with a more balanced class distribution. Due to changes in video availability, we downloaded 1,772,023 AudioSet-2M training, 18,691 AudioSet-20K training, and 17,249 evaluation samples, respectively. VGGSound Chen et al. (2020) is a collection of 200K 10-second YouTube video clips annotated with 309 classes. We download 183,727 training and 15,446 test samples. We only use the labels in the fine-tuning stage to make our pretraining pipeline fully self-supervised. Compared with AudioSet, one advantage of VGGSound is that the sound source is always visually evident within the video clip, which is done by filtering the videos with a pretrained vision classifier. As discussed in Li et al. (2022), different versions of dynamic datasets might cause a performance difference, to improve the reproducibility of this work, we release the training and test samples ids at `https://github.com/yuangongnd/cav-mae`.

## B  TRAINING DETAILS

Our training hyper-parameters are listed in Table 4. Most of our experiments are run on 4×NVIDIA GTX Titan X Pascal GPUs with 12GB memory, only the scaled-up CAV-MAE$^{\texttt{Scale+}}$ is pretrained on 4×NVIDIA RTX A5000 GPUs with 24GB memory, making our result easier to reproduce with reasonable resources. Pretraining CAV-MAE takes about one week with 4 GPUs.

Our model has a similar size with "base" MAE models, i.e., the full encoder and decoder model has ~190M parameters (due to two modal-specific branches); the encoder used for audio-visual downstream task is ~160M parameters; the encoder used for single-modal downstream task is ~85M parameters.

Table 4: Our pre-training and fine-tuning hyperparameters.

| | Pretraining | | Finetuning | | |
|---|---|---|---|---|---|
| | `CAV-MAE`[Scale+] | All Other Models | All Models | | |
| Dataset | AS-2M | AS-2M | AS-20K | AS-2M | VGG |
| Optimizer | Adam, weight decay=5e-7, betas=(0.95, 0.999) | | | | |
| Backbone learning rate | 1e-4 | 5e-5 | 5e-5 | 1e-5 | 1e-4 |
| Classification head LR | - | - | 5e-2 | 5e-4 | 1e-3 |
| LR decay start epoch | 10 | 10 | 5 | 2 | 2 |
| LR decay rate | 0.5 | 0.5 | 0.5 | 0.5 | 0.5 |
| LR decay step | 5 | 5 | 1 | 1 | 1 |
| Epochs | 25 | 12 | 15 | 10 | 10 |
| Batch size | 4×27 | 4×12 | 36 | 48 | 48 |
| GPUs | 4 A5000 | 4 Titan X Pascal | | | |
| Class Balance Sampling | No | No | No | Yes | Yes |
| Mixup | No | No | Yes | Yes | Yes |
| Random Time Shifting | Yes | Yes | Yes | Yes | Yes |
| Loss Function | - | | BCE | BCE | CE |
| Weight Averaging | No | No | Yes | Yes | Yes |
| Ensemble | No | No | No | No | No |
| Input Norm Mean | -5.081 | -5.081 | -5.081 | -5.081 | -5.081 |
| Input Norm STD | 4.485 | 4.485 | 4.485 | 4.485 | 4.485 |

## C  AUDIO-VISUAL ACTION RECOGNITION EXPERIMENTS

In addition to the audio-visual event classification task on AudioSet and VGGSound, we also test our models on the audio-visual action recognition tasks. One problem with existing audio-visual action recognition datasets is they are usually visual-heavy and dominated by the performance of the visual branch. Therefore, to test our audio-visual model, we choose to conduct experiments on Kinetics-Sounds (Arandjelovic & Zisserman, 2017), a subset of Kinetics-400 dataset (Kay et al., 2017) with $32^2$ human action classes that have been chosen to be potentially manifested visually and aurally.

We conduct two experiments on Kinetics-Sounds:

First, we pretrain and fine-tune CAV, AV-MAE, and CAV-MAE using the Kinetics-Sounds training set and report the Top-1 accuracy on the Kinetics-Sounds validation set (i.e., no AudioSet pretraining). This is to check if CAV-MAE still outperforms its counterparts on the audio-visual action recognition task. As shown in Table 5, the conclusion on Kinetics-Sounds is consistent with that on AudioSet and VGGSound, i.e., CAV-MAE performs better than both CAV and AV-MAE.

Second, we compare CAV-MAE models with SOTA MBT model (Nagrani et al., 2021) following the protocol of MBT. Specifically, we train the model on Kinetics-400 (K400) dataset and report the top-1 accuracy on Kinetics-Sounds. We find the label set used impacts the accuracy and this setting is not clear in the MBT paper. Therefore, we report the results on both the Kinetics-400 label set (i.e., not restrict predictions in 32 Kinetics-Sounds classes) and the Kinetics-Sounds label set (i.e., restrict predictions in 32 Kinetics-Sounds classes). As shown in Table 6, our CAV-MAE matches MBT on Kinetics-Sounds. Please note that our CAV-MAE model is trained in a fully self-supervised manner while MBT uses supervised ImageNet pretrained weights. For the difference between ImageNet

---

[2]The original Kinetics-Sounds dataset consists of 34 classes with an early version of Kinetics-400 label set. We contact the authors and use the 32-class label set defined in (Xiao et al., 2020) for our experiments.

supervised learning (SL) model and self-supervised learning (SSL) model initialization, please see Table 11.

Table 5: Comparison of CAV, AV-MAE, and CAV-MAE models on Kinetics-Sounds. For each model, we pretrain and fine-tune it using the Kinetics-Sounds training set and report the Top-1 accuracy on the Kinetics-Sounds validation set. The conclusion is consistent with our AudioSet and VGGSound experiments that CAV-MAE outperforms both CAV and AV-MAE.

|         | Kinetics-Sounds Accuracy |
|---------|--------------------------|
| CAV     | 86.2 |
| AV-MAE  | 88.0 |
| CAV-MAE | 88.9 |

Table 6: Comparison of CAV-MAE models with SOTA MBT model (Nagrani et al., 2021) on Kinetics-Sounds. Following the protocol of MBT, we train the model on Kinetics-400 (K400) dataset and report the top-1 accuracy on Kinetics-Sounds. We report the results on both the Kinetics-400 label set (i.e., not restrict predictions in 32 Kinetics-Sounds classes) and the Kinetics-Sounds label set (i.e., restrict predictions in 32 Kinetics-Sounds classes). Our CAV-MAE matches or outperforms MBT on Kinetics-Sounds with a fully self-supervised learning (SSL) setting.

|         | Out-of-Domain Pretrain | In-Domain Training | K400 Label Set | Kinetics-Sounds Label Set |
|---------|------------------------|--------------------|----------------|---------------------------|
| MBT     | ImageNet SL            | K400 SL            |                | 85.0 |
| CAV-MAE | No                     | K400 SSL + SL      | 70.6           | 83.3 |
| CAV-MAE | ImageNet SSL           | K400 SSL + SL      | 83.2           | 90.6 |
| CAV-MAE | ImageNet + AudioSet SSL | K400 SSL + SL     | 85.0           | 90.9 |

# D  ADDITIONAL AUDIO-VISUAL RETRIEVAL RESULTS.

## D.1  AUDIO TO VISUAL RETRIEVAL RESULTS ON AUDIOSET AND VGGSOUND

We show audio to visual retrieval results on AudioSet and VGGSound (zero-shot) in Table 7.

Table 7: Audio to visual retrieval results on AudioSet and VGGSound.

| Audio→Visual Retrieval | AudioSet Eval Subset | | | VGGSound Eval Subset | | |
|------------------------|------|------|------|------|------|------|
|                        | R@1  | R@5  | R@10 | R@1  | R@5  | R@10 |
| *Audio-Visual Models with Only MDM Loss* | | | | | | |
| Vanilla AV-MAE         | 0.2  | 0.4  | 0.9  | 0.0  | 0.4  | 0.8 |
| AV-MAE                 | 0.2  | 0.4  | 0.9  | 0.0  | 0.2  | 0.6 |
| *Audio-Visual Models with Only Contrastive Loss* | | | | | | |
| CAV, $\lambda_c = 0.1$  | 15.5 | 32.7 | 42.8 | 12.4 | 33.2 | 44.7 |
| CAV, $\lambda_c = 0.01$ | 11.5 | 27.5 | 36.5 | 10.0 | 25.6 | 36.9 |
| *Constrastive Audio-Visual Masked Auto-Encoders* | | | | | | |
| CAV-MAE, $\lambda_c = 0.1$  | 13.5 | 32.5 | 43.2 | 12.1 | 31.6 | 42.4 |
| CAV-MAE, $\lambda_c = 0.01$ | 9.5  | 22.6 | 32.4 | 8.3  | 23.8 | 32.4 |
| CAV-MAE(Scale), C=0.01 | 15.1 | 34.0 | 43.0 | 12.8 | 30.4 | 40.3 |

## D.2  VGGSOUND RETRIEVAL SAMPLES

We show bi-directional zero-shot VGGSound retrieval samples in Figure 7 and Figure 8.

Table 8: Audio-visual bi-directional retrieval results on MSR-VTT dataset. All models, including the baseline models, are initialized with ImageNet weights and trained with *only* MSR-VTT data. Our CAV and CAV-MAE models outperform existing methods in both directions. In addition, comparing CAV and CAV-MAE, we again find the MAE training objective does not hurt, or even improve the retrieval performance.

| | Audio→Visual | | | Visual→Audio | | |
|---|---|---|---|---|---|---|
| | R@1 | R@5 | R@10 | R@1 | R@5 | R@10 |
| Random | 0.1 | 0.5 | 1 | 0.1 | 0.5 | 1 |
| Boggust et al. (2019) | 1.0 | 3.8 | 7.1 | 1.8 | 4.5 | 8.1 |
| Arandjelovic & Zisserman (2018) | 1.3 | 4.3 | 8.2 | 0.3 | 2.5 | 6.6 |
| AVLnet (Rouditchenko et al., 2021) | 0.9 | 5.0 | 9.0 | 0.8 | 4.6 | 8.1 |
| CAV, $\lambda_c = 0.1$ | 0.2 | 4.8 | 10.4 | 1.9 | **9.6** | **14.9** |
| CAV-MAE, $\lambda_c = 0.1$ | **1.5** | **8.0** | **12.4** | **2.6** | 9.2 | 13.1 |

Table 9: Zero-shot audio-visual bi-directional retrieval results on MSR-VTT dataset. Existing methods are trained with the 100M HowTo100M dataset, while our models are only trained with the 2M AudioSet dataset. With less than 2% of pretraining data, our CAV-MAE model achieves similar results for visual-audio retrieval performance with existing methods. Again, CAV-MAE models have similar or better results compared with CAV models when $\lambda_c$ is the same.

| | Pretrain Dataset | Audio→Visual | | | Visual→Audio | | |
|---|---|---|---|---|---|---|---|
| | | R@1 | R@5 | R@10 | R@1 | R@5 | R@10 |
| Boggust et al. (2019) | HowTo100M | 7.6 | 21.1 | 28.3 | 9.3 | 20.7 | 28.8 |
| Arandjelovic & Zisserman (2018) | HowTo100M | 12.6 | 26.3 | 33.7 | 11.9 | 25.9 | 34.7 |
| AVLnet (Rouditchenko et al., 2021) | HowTo100M | 17.8 | 35.5 | 43.6 | 17.2 | 26.6 | 46.6 |
| CAV, $\lambda_c = 0.1$ | AudioSet-2M | 6.2 | 17.9 | 26.1 | 10.5 | 25.2 | 35.5 |
| CAV-MAE, $\lambda_c = 0.1$ | AudioSet-2M | 7.0 | 18.7 | 28.6 | 10.0 | 26.5 | 38.0 |
| CAV, $\lambda_c = 0.01$ | AudioSet-2M | 4.2 | 14.8 | 22.7 | 6.9 | 22.5 | 33.1 |
| CAV-MAE, $\lambda_c = 0.01$ | AudioSet-2M | 4.9 | 14.6 | 21.9 | 8.3 | 23.5 | 35.0 |
| CAV-MAE$^{\text{Scale+}}$, $\lambda_c = 0.01$ | AudioSet-2M | 7.6 | 19.8 | 30.2 | 13.3 | 29.0 | 40.5 |

### D.3 MSR-VTT Dataset Retrieval Experiments

We also conduct audio-visual retrieval experiments on MSR-VTT (Xu et al., 2016) and compare our models with existing works. Specifically, we conduct two sets of experiments.

First, we train CAV and CAV-MAE models on the MSR-VTT training set and evaluate them on the MSR-VTT test set. Note the models are not pretrained on AudioSet. We then compare the retrieval performance with existing works in the same training setting. As shown in Table 8, our CAV and CAV-MAE models outperform existing methods in both directions. In addition, comparing CAV and CAV-MAE, we again find the MAE training objective does not hurt, or even improve the retrieval performance.

Second, we conduct a *zero-shot* retrieval experiment on MSR-VTT. Specifically, we take the AudioSet pretrained models and directly evaluate them on the MSR-VTT test set. The MSR-VTT training set is not used. We then compare our models with existing models. As shown in Table 9, our CAV-MAE model achieves similar results for visual-audio retrieval performance with existing methods but worse for the audio-visual direction. However, existing methods are trained with the 100M HowTo100M dataset, while our models are only trained with the 2M AudioSet dataset. With less than 2% of training data, our CAV-MAE model achieves similar results for visual-audio retrieval performance with existing methods. Again, CAV-MAE models still have similar or better results compared with CAV models when $\lambda_c$ is the same, demonstrating the MAE and contrastive objective do not conflict.

# E  IMPACT OF MODEL INITIALIZATION

Existing audio-visual models typically use (supervised) ImageNet pretrained weights to initialize the model. Throughout the paper, we always initialize our models (including CAV, AV-MAE, and CAV-MAE) with self-supervised ImageNet pretrained weights. Specifically, we use the weight from the original vision MAE model (He et al., 2022) (Weights from `https://github.com/facebookresearch/mae`) with only self-supervised learning (SSL) pretraining for *all* audio, visual, and joint encoder and the decoder. This is implemented by duplicating the weights of MAE encoder layer 1-11 for the audio and visual encoder, respectively, and the weights of MAE encoder layer 12 for the joint encoder.

How important is this initialization? We conduct experiments with various model initialization and pretraining settings. As shown in Table 10, we find that ImageNet initialization always leads to a performance improvement, no matter in fine-tuning or linear probing test, and such improvement decreases with a larger in-domain pretraining dataset, e.g., without ImageNet initialization, CAV-MAE performs just 1.0% mAP lower on AudioSet-2M. Therefore, ImageNet initialization is not an indispensable component of the proposed CAV-MAE pretraining framework.

Finally, we quantify the difference between initialing the model with ImageNet SSL pretrained weights and ImageNet SL pretrained weights on the downstream task. As shown in Table 11, on AudioSet-20K, using SL weights leads to a 3.7% improvement over using SSL weights in the fine-tuning setting (but interestingly, in the linear probing setting, SL weights lead to worse results). Therefore, directly comparing our fully self-supervised model with existing models with a supervised pretraining component is not exactly fair.

Table 10: CAV-MAE model performance with various model initialization and pretraining settings on AudioSet-20K, VGGSound, and AudioSet-2M. We report both end-to-end fine-tuning and linear probing results. Initializing CAV-MAE with ImageNet pretrained weights consistently improves the model performance, but is not an indispensable component. Without ImageNet initialization, CAV-MAE performs just 1.0% mAP lower on AudioSet-2M.

| Settings | | AudioSet-20K | | VGGSound (200K) | | AudioSet-2M |
| --- | --- | --- | --- | --- | --- | --- |
| ImageNet Initialization | AudioSet Pretraining | Fine Tuning | Linear Probing | Fine Tuning | Linear Probing | Fine Tuning |
| No | No | 8.0 | 2.4 | 42.4 | 10.3 | 33.5 |
| SSL | No | 25.6 | 10.3 | 62.1 | 34.3 | 47.3 |
| No | SSL | 37.3 | 29.1 | 62.7 | 53.0 | 49.5 |
| **SSL** | **SSL** | **40.6** | **29.8** | **65.4** | **54.2** | **50.5** |

Table 11: Most existing audio-visual models initialize their weights with ImageNet supervise pretrained weights (e.g., Nagrani et al. (2021); Rouditchenko et al. (2021)) while we intend to build a fully self-supervised model to avoid using any labels. Compare the AudioSet-20K performance of models initialized with ImageNet supervised pretrained (SL) weights and ImageNet self-supervised pretrained (SSL) weights. The SL weights and SSL weights are from the original MAE models with and without supervised ImageNet finetuning (He et al., 2022), respectively. Since the SL weights only contain weights of the MAE encoder part and cannot be used for further SSL pretraining. We directly fine-tune/linear probe the two models on AudioSet-20K (i.e., no in-domain pretraining) and report the results to make a fair comparison. We observe that initialing the model SL weights leads to a noticeable advantage for fine-tuning, showing the ImageNet labels are still very valuable supervision signals. This also indicates that directly comparing our fully self-supervised model with existing models with a supervised pretraining component is not exactly fair.

| Initialization Weight | AudioSet-20K | |
| --- | --- | --- |
| | Fine Tuning | Linear Probing |
| ImageNet SSL | 25.6 | 10.3 |
| ImageNet SL | 29.3 | 7.2 |

## F  IMPACT OF MASKING STRATEGY AND MASKING RATIO

### F.1  IMPACT OF TRAINING MASKING RATIO

Throughout the paper, we use a 75% masking ratio for both audio and visual input. This is mainly due to many previous MAE works reporting a masking ratio ~75% is appropriate for both audio and visual input He et al. (2022); Baade et al. (2022); Huang et al. (2022a); Niizumi et al. (2022). However, it is unclear if such a high masking ratio is also appropriate for the contrastive objective. In particular, aggressive augmentation is not commonly used in audio-visual contrastive learning. Therefore, we conduct experiments to check the impact of the training masking ratio on the audio-visual joint event classification task and the audio-visual retrieval task.

For the audio-visual joint event classification task, as shown in Table 12, we find the CAV model does perform slightly better with a smaller masking ratio (50%), but the difference is minor. When the masking ratio is 75%, CAV still performs well. This shows the audio-visual joint classification task is not sensitive to the masking ratio.

For the audio-visual retrieval task, as shown in Table 13, we find that the audio-visual retrieval performance decreases with a higher masking ratio, particularly when the masking ratio is very high. If audio-visual retrieval is the main task of interest, a lower masking ratio should be used in training, which does not hurt the audio-visual joint event classification task, but requires more computation. In Section 5 and Appendix D, we show CAV-MAE is already a strong audio-visual retrieval model when the masking ratio is 75%, the performance can be further improved by lowering the masking ratio. Note this result does not conflict with the fact that the reconstruction objective does not hurt, and in many cases, improves the retrieval performance.

Table 12: Audio-visual joint event classification performance of CAV, AV-MAE, and CAV-MAE as a function of masking ratio on AudioSet-20K and VGGSound. All models are pretrained with uniform unstructured masking. We find the contrastive learning model CAV performs slightly better with a lower masking ratio while the AV-MAE model performs best with ~75% masking ratio. These results show that a 65%~75% masking ratio works well for both contrastive learning and masked data modeling frameworks for the downstream audio-visual joint event classification task.

| Masking Ratio | AudioSet-20K | | | VGGSound | | |
|---|---|---|---|---|---|---|
| | CAV | AV-MAE | CAV-MAE | CAV | AV-MAE | CAV-MAE |
| 0.50 | 38.5 | 37.2 | 41.2 | 64.3 | 63.7 | 65.1 |
| 0.65 | 38.6 | 37.9 | 41.3 | 64.0 | 64.1 | 64.9 |
| 0.75 | 38.5 | 37.4 | 40.5 | 64.1 | 64.1 | 65.4 |
| 0.85 | 38.1 | 37.3 | 40.3 | 63.3 | 64.2 | 64.7 |

Table 13: Zero-shot audio-visual retrieval performance of CAV-MAE ($\lambda_c = 0.01$) as a function of masking ratio on VGGSound evaluation subset. All models are pretrained with uniform unstructured masking. The audio-visual retrieval performance decreases with a higher masking ratio.

| Masking Ratio | Audio→Visual | | | Visual→Audio | | |
|---|---|---|---|---|---|---|
| | R@1 | R@5 | R@10 | R@1 | R@5 | R@10 |
| 0.50 | 14.8 | 34.9 | 45.4 | 15.2 | 36.9 | 48.3 |
| 0.65 | 11.3 | 28.9 | 39.0 | 14.2 | 33.5 | 45.2 |
| 0.75 | 8.3 | 23.8 | 32.4 | 12.5 | 28.6 | 39.1 |
| 0.85 | 5.6 | 16.0 | 22.7 | 8.7 | 22.6 | 30.6 |

### F.2  IMPACT OF AUDIO TRAINING MASKING STRATEGY

Another key design of masking is the masking strategy. Throughout the paper, we use a uniform, unstructured masking strategy for both audio and visual input. However, unlike visual modalities, the two dimensions of audio spectrograms are heterogeneous. In this section, we explore the impact

of masking strategies for audio input. Specifically, we apply time, frequency, and time-frequency masking strategies (depicted in Figure 3) and compare them with the uniform unstructured masking strategy (i.e., uniform masking).

For the audio-visual joint event classification task, as shown in Table 14, we find that all four training masking strategies lead to similar performance when the training masking ratio is 75%. However, as we show in Figure 5, structured masking strategies make reconstruction more challenging. Therefore, we also pretrain a CAV-MAE model trained with time-frequency masking at a lower masking ratio of 50%, which shows slightly better performance on both AudioSet-20K and VGGSound. In general, the audio-visual joint classification task is not sensitive to the masking strategy.

For the audio-visual retrieval task, as shown in Table 15, with the same 75% masking ratio, different masking strategies lead to noticeably different retrieval performance. Frequency and time-frequency masking leads to the best retrieval performance while unstructured uniform masking actually leads to the worst retrieval performance. In Section 5 and Appendix D, we show CAV-MAE is already a strong audio-visual retrieval model when uniform masking is used, the performance can be further improved by using a structured masking strategy, which also does not hurt the audio-visual joint event classification.

To summarize, we find both the masking ratio and masking strategy have a minor impact on the downstream audio-visual joint event classification task, but have a noticeable impact on the audio-visual retrieval task. Specifically, there exist masking strategies that lead to better retrieval performance than the default 75% uniform masking strategy. Finally, we also notice the training masking strategy impacts the model reconstruction ability, which is discussed in Section H.2.

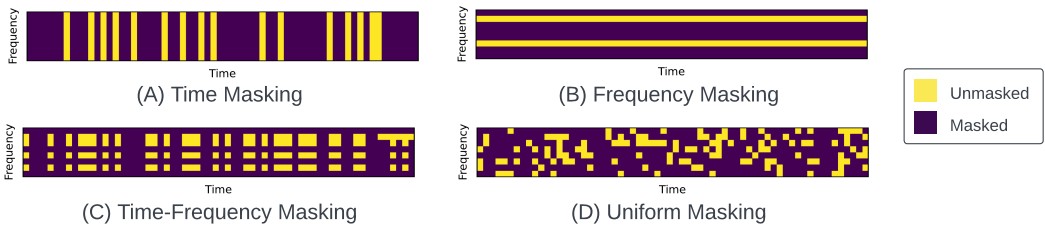

Figure 3: Illustration of various masking strategies. We use uniform unstructured masking throughout the paper except in Section F.

Table 14: Audio-visual joint event classification performance of CAV-MAE as a function of training masking strategy and ratio on AudioSet-20K and VGGSound. We find that all four training masking strategies lead to similar performance when the training masking ratio is 75%. However, as we show in Figure 5, structured masking strategies make reconstruction more challenging. Therefore, we also pretrain a CAV-MAE model trained with time-frequency masking at a lower masking ratio of 50%, which shows slightly better performance on both AudioSet-20K and VGGSound.

| Masking | | AudioSet-20K | VGGSound |
|---|---|---|---|
| Strategy | Ratio | | |
| Uniform | 0.75 | 40.5 | 65.4 |
| Time | 0.75 | 40.6 | 64.6 |
| Frequency | 0.75 | 40.4 | 65.0 |
| Time-Frequency | 0.75 | 40.7 | 64.8 |
| Time-Frequency | 0.50 | 41.2 | 65.6 |

Table 15: Zero-shot audio-visual retrieval performance of CAV-MAE ($\lambda_c = 0.01$) as a function of training masking strategy on VGGSound evaluation subset. All models are trained with a masking ratio of 75% on AudioSet. The masking strategy has a noticeable impact on retrieval performance.

| Masking Ratio | Audio→Visual | | | Visual→Audio | | |
|---|---|---|---|---|---|---|
| | R@1 | R@5 | R@10 | R@1 | R@5 | R@10 |
| Uniform | 8.3 | 23.8 | 32.4 | 12.5 | 28.6 | 39.1 |
| Time | 10.1 | 26.0 | 35.5 | 12.5 | 30.2 | 40.3 |
| Frequency | 11.7 | 31.9 | 42.7 | 13.7 | 34.7 | 45.8 |
| Time-Frequency | 13.1 | 32.8 | 42.1 | 14.7 | 36.4 | 47.3 |

## G    IMPACT OF THE NUMBER OF FRAMES USED

In the paper, we sample 10 frames for each 10-second video clip (1 FPS). How does the frame rate impact the performance? As shown in Figure 4, on all Kinetics-Sounds, AudioSet-20K, and VGGSound, higher FPS consistently improves the downstream classification performance, however, the improvement saturates with the increasing of frames.

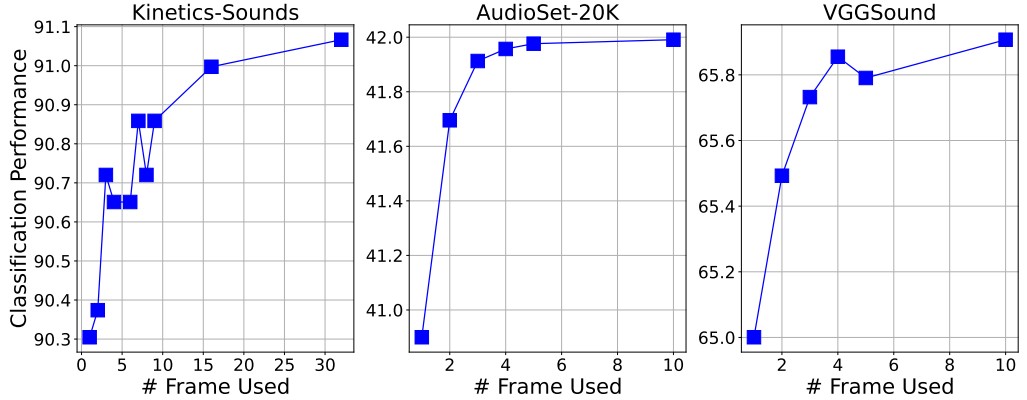

Figure 4: Classification performance as a function of the number of frames used on Kinetics-Sounds (left), AudioSet-20K (middle), and VGGSound (right). Frames are uniformly sampled from each video clip. The performance consistently improves with more frames being used, but the improvement saturates with the increase of frames.

## H    CAV-MAE RECONSTRUCTION RESULTS

### H.1    AUDIO-VISUAL RECONSTRUCTION SAMPLES

We show the CAV-MAE reconstruction samples in Figure 9, 10, and 11. All samples are from VGGSound, a different dataset from the pretraining set. The CAV-MAE model is trained with a 75% masking ratio *without* target normalization. As shown in Table 2d., it has a similar performance to the default model with target normalization. CAV-MAE has strong reconstruction ability even if the masking ratio goes to 90%, which makes it potentially can be used for in-painting and enhancement tasks. All inference masks are sampled uniformly (i.e., unstructured masking).

### H.2    AUDIO SPECTROGRAM RECONSTRUCTION UNDER VARIOUS INFERENCE MASKING SETTINGS

Besides uniform masking samples shown in the previous section, we also show the audio spectrogram reconstruction samples under various structured inference masking settings in Figure 12 (75% masking ratio) and Figure 13 (90% masking ratio). We find structured masking is more challenging

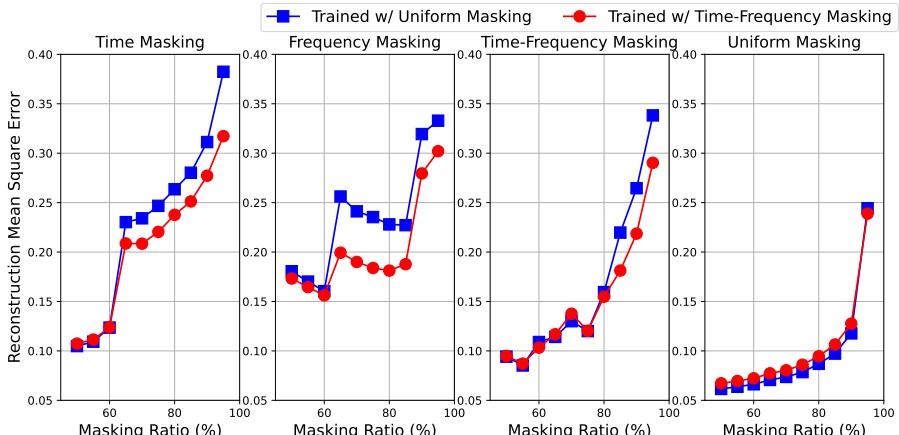

Figure 5: Audio spectrogram reconstruction mean squared error (MSE) as a function of masking ratio under various inference masking settings (from left to right: time masking, frequency masking, time-frequency masking, and uniform unstructured masking). We compare a CAV-MAE model trained with uniform masking (blue) and a CAV-MAE model trained with time-frequency masking (red). Both models are trained with a 75% masking ratio. Key findings are as follows: 1) Even for the same masking ratio, the reconstruction hardness is different for each masking strategy. On average, time masking is the most difficult, followed by frequency masking, time-frequency masking, and uniform unstructured masking. This indicates that CAV-MAE models require *local* information for the reconstruction task. However, for each specific spectrogram, the order of difficulty varies (see Figure 12 and 13). Second, the CAV-MAE model trained with time-frequency masking generally performs better than its counterpart trained with uniform masking in audio spectrogram reconstruction, particularly for the time masking and frequency masking settings, showing it is stronger in leveraging *global* information. This indicates different training masking strategies do impact the properties of the model.

for reconstruction as the mean squared errors are generally higher. On average, time masking is the most difficult, followed by frequency masking, time-frequency masking, and uniform unstructured masking. This also indicates that the model leverages *local* neighboring unmasked part information to infer the masked part. When an entire time or frequency span is masked, the model is harder to reconstruct (this is quantified in Figure 5).

Finally, in Figure 12 and Figure 13, we also compare the reconstruction ability of a CAV-MAE model trained with uniform, unstructured masking strategy and a CAV-MAE model trained with time-frequency masking strategy (both with 75% masking ratio). We quantify the difference in Figure 5. Interestingly, we find the CAV-MAE model trained with time-frequency masking generally performs better than its counterpart trained with uniform masking in audio spectrogram reconstruction, particularly for the time masking and frequency masking settings, showing it is stronger in leveraging *global* information. This indicates different training masking strategies do impact the properties of the model. While the training masking strategy only minorly impacts the downstream classification task, it has a relatively large impact on reconstruction.

## I CAV-MAE VISUAL SOUND SOURCE LOCALIZATION RESULTS

We evaluate the capability of CAV-MAE (uniform masking, masking ratio = 75%, $\lambda_c$=0.01) on the visual sound source localization task with a basic similarity-based method. Specifically, for each audio-image pair, we mean pool the representations of all audio tokens as the clip-level audio representation, and then calculate the cosine similarity between the clip-level audio representation with all patch-level image representations as the visual sound source localization heat map. In general, we find the CAV-MAE model is not a strong visual sound source localization model though its audio-visual retrieval performance is good. In Figure 6, we show a successful sample (left) and a failed sample (right). In some cases, CAV-MAE localizes the sound to the background instead of

the main sound source object. We hypothesize that it is due to the *masked* contrastive learning objective. During the training process, the model needs to match positive audio-visual pairs even when both modalities are heavily masked, in some situations, the main sound source could be completely masked, the model thus learns to leverage the context information for the matching, which may hurt its performance on the visual sound source localization task.

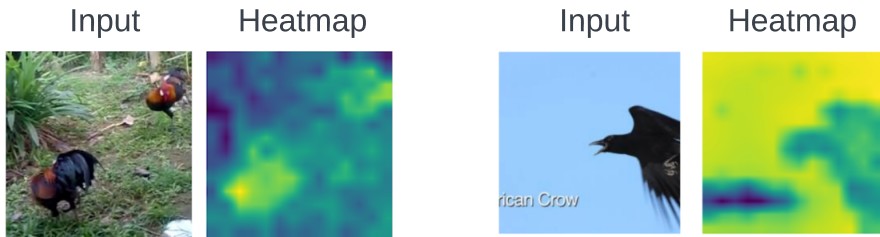

Figure 6: A successful sample (left) and a failed sample (right) of CAV-MAE on the visual sound source localization task. In some cases, CAV-MAE localizes the sound to the background instead of the main sound source object.

## J Impact of the Audio-Visual Pairing Information in Training Dataset

Even without a contrastive objective, AV-MAE allows the model to reconstruct one modality based on the information of another modality, which theoretically allows the model to learn audio-visual correlation. However, without an explicit objective of encouraging paired audio-visual correspondence, to which extent the AV-MAE leverages the audio-visual pairing information is unknown. In this section, we evaluate this by the following experiment: we break the original audio-visual pairs of the training set and conduct a random shuffle (i.e., randomly match audio and visual samples in the dataset), which removes most of the audio-visual pairing information in the training data. We train the CAV, AV-MAE, and CAV-MAE models with the shuffled training dataset, and then finetune these models on the audio-visual joint event classification task with original unshuffled downstream datasets. As shown in Table 16, we find 1) the CAV model that solely relies on the audio-visual pairing information has a significant performance drop when the training dataset is shuffled; 2) the AV-MAE model is almost not impacted by the training set shuffle, indicating it is weak at leveraging audio-visual pairing information in the training set and mostly relies on single-modality information; 3) CAV-MAE performs almost the same with AV-MAE with the shuffled training set, but noticeably better with the original training set. These findings again justify the main point of this paper that contrastive and reconstruction objectives are most effective when they are combined together. When only the contrastive objective is used, the model performs worse and is less robust to the noise in the training set. When only the reconstruction objective is used, the model does not effectively leverage the audio-visual pair information.

Table 16: Comparing the audio-visual joint event classification performance of models trained with AudioSet with original audio-visual pairs and AudioSet with randomly shuffled audio-visual pairs.

| Training Data | AudioSet-20K | | | VGGSound | | |
|---|---|---|---|---|---|---|
| | CAV | AV-MAE | CAV-MAE | CAV | AV-MAE | CAV-MAE |
| Shuffled AudioSet-2M | 1.25 | 37.4 | 37.4 | 7.28 | 64.1 | 64.1 |
| Original AudioSet-2M | 38.5 | 37.4 | 40.5 | 64.1 | 64.1 | 65.4 |

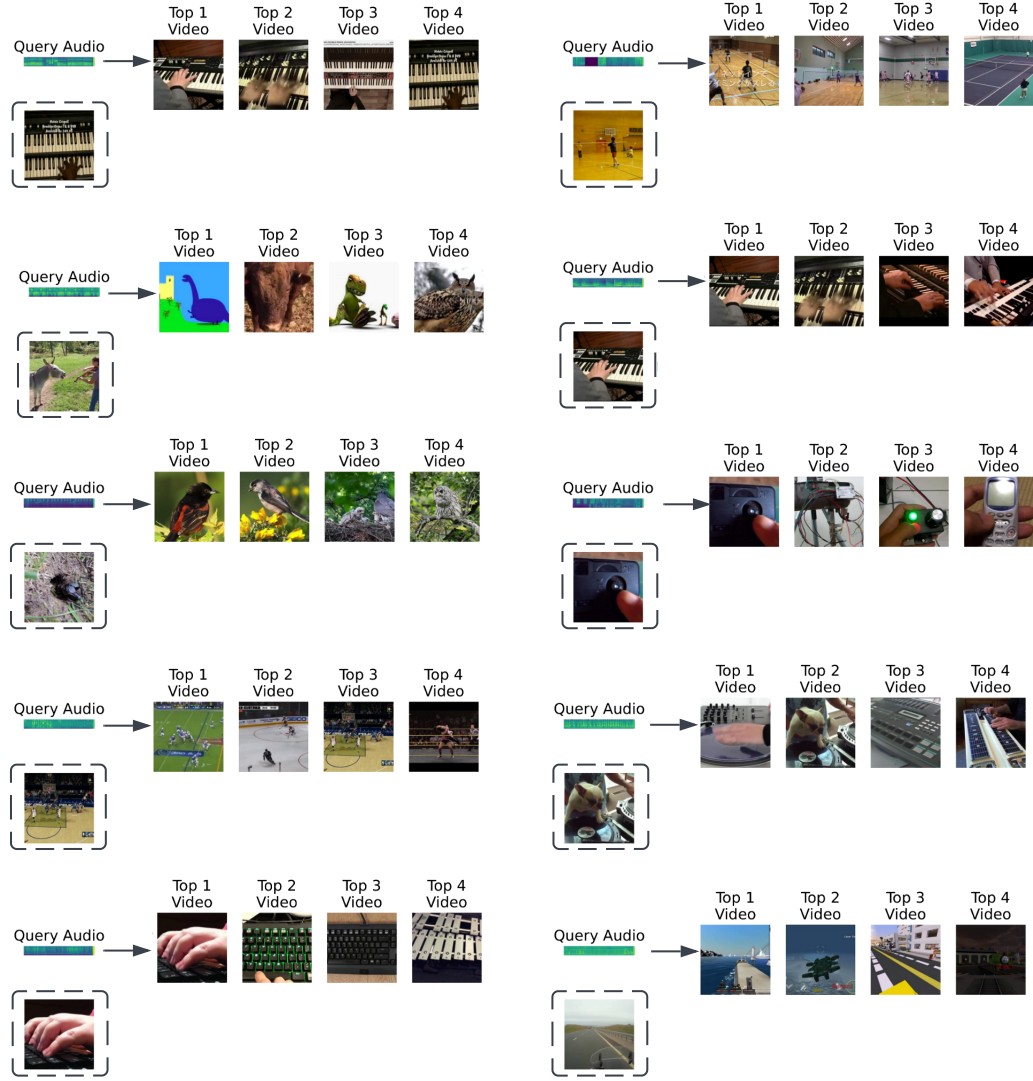

Figure 7: Zero-shot audio to image retrieval results on VGGSound. Since the spectrograms are hard to read, we show their paired images in the dashed boxes for visualization purposes, only audios are used as queries.

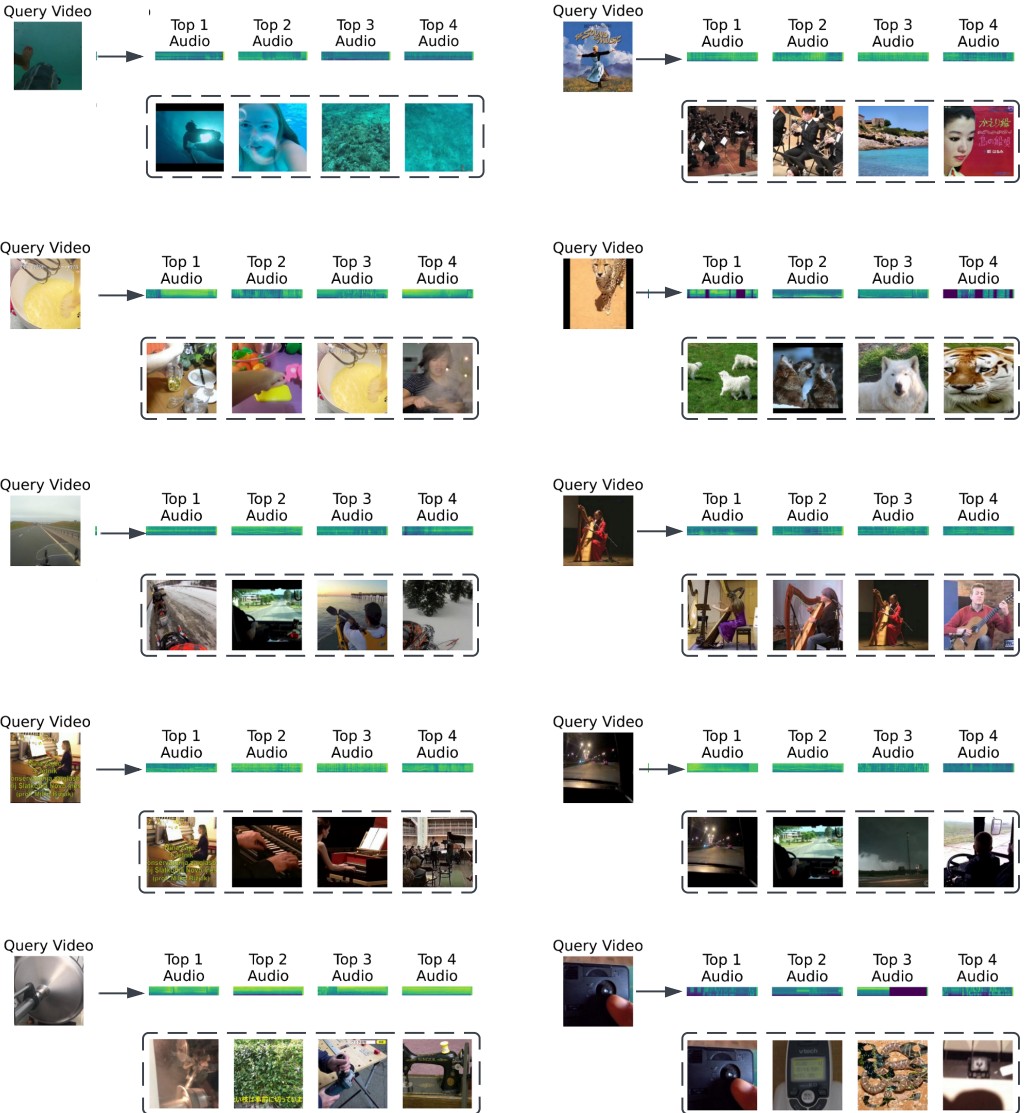

Figure 8: Zero-shot image to audio retrieval results on VGGSound. Since the spectrograms are hard to read, we show their paired images in the dashed boxes for visualization purposes, only audios are used as keys.

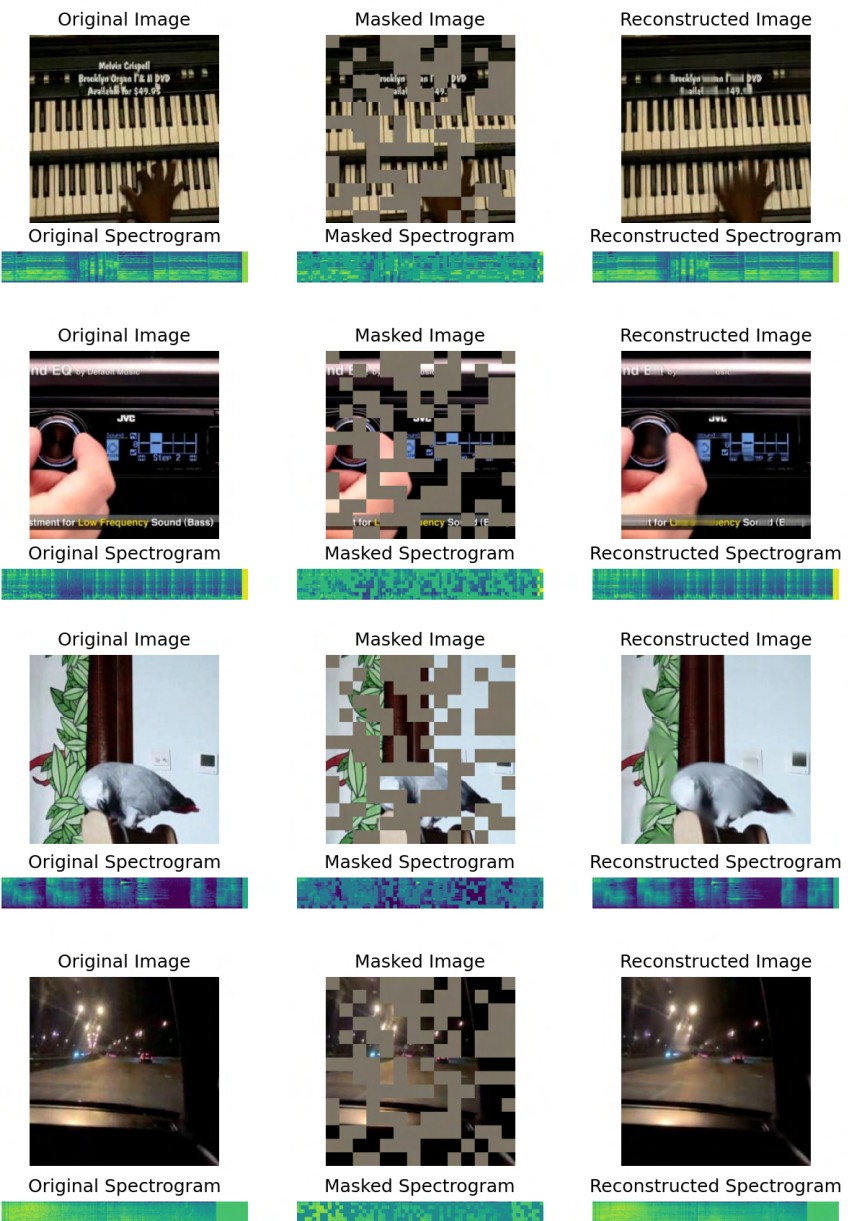

Figure 9: CAV-MAE reconstruction samples when 50% of the input is masked. Samples are from VGGSound, a different dataset from the pretraining dataset. The model is pretrained on AudioSet with a 75% masking ratio without target normalization.

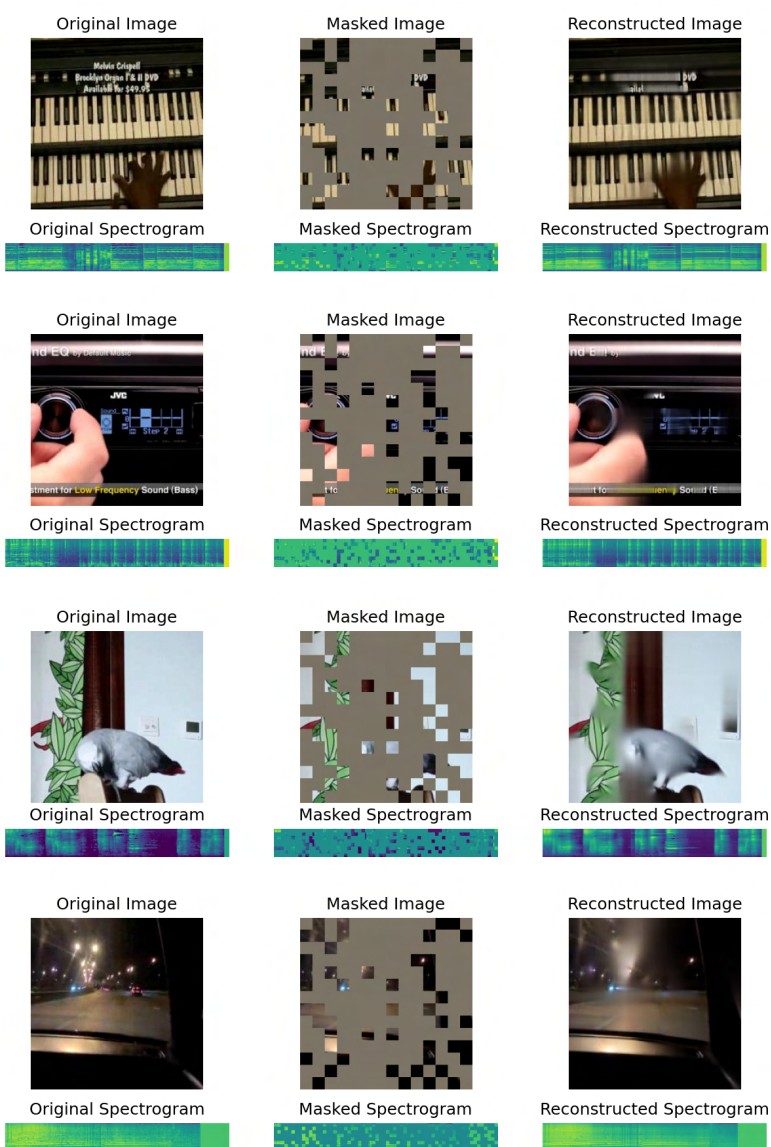

Figure 10: CAV-MAE reconstruction samples when 75% of the input is masked. Samples are from VGGSound, a different dataset from the pretraining dataset. The model is pretrained on AudioSet with a 75% masking ratio without target normalization.

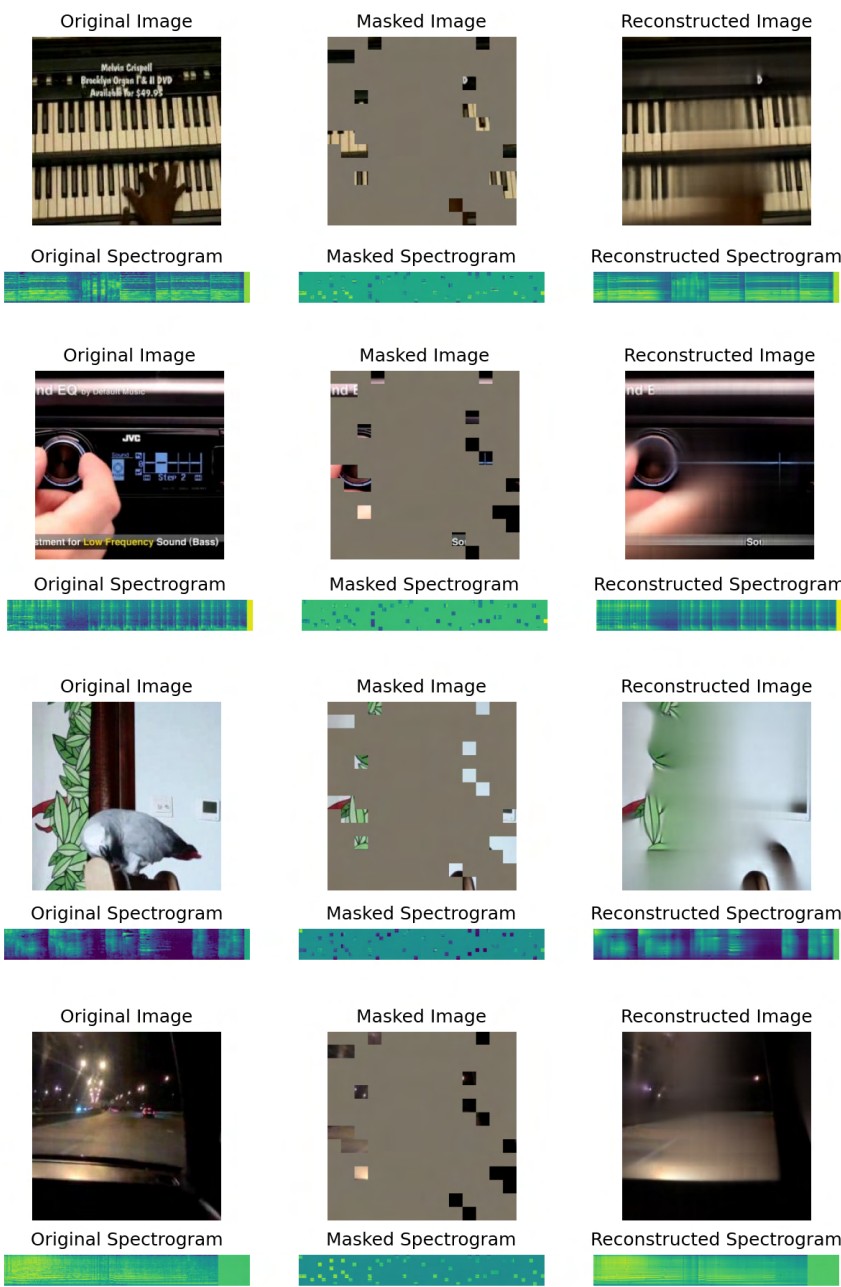

Figure 11: CAV-MAE reconstruction samples when 90% of the input is masked. Samples are from VGGSound, a different dataset from the pretraining dataset. The model is pretrained on AudioSet with a 75% masking ratio without target normalization.

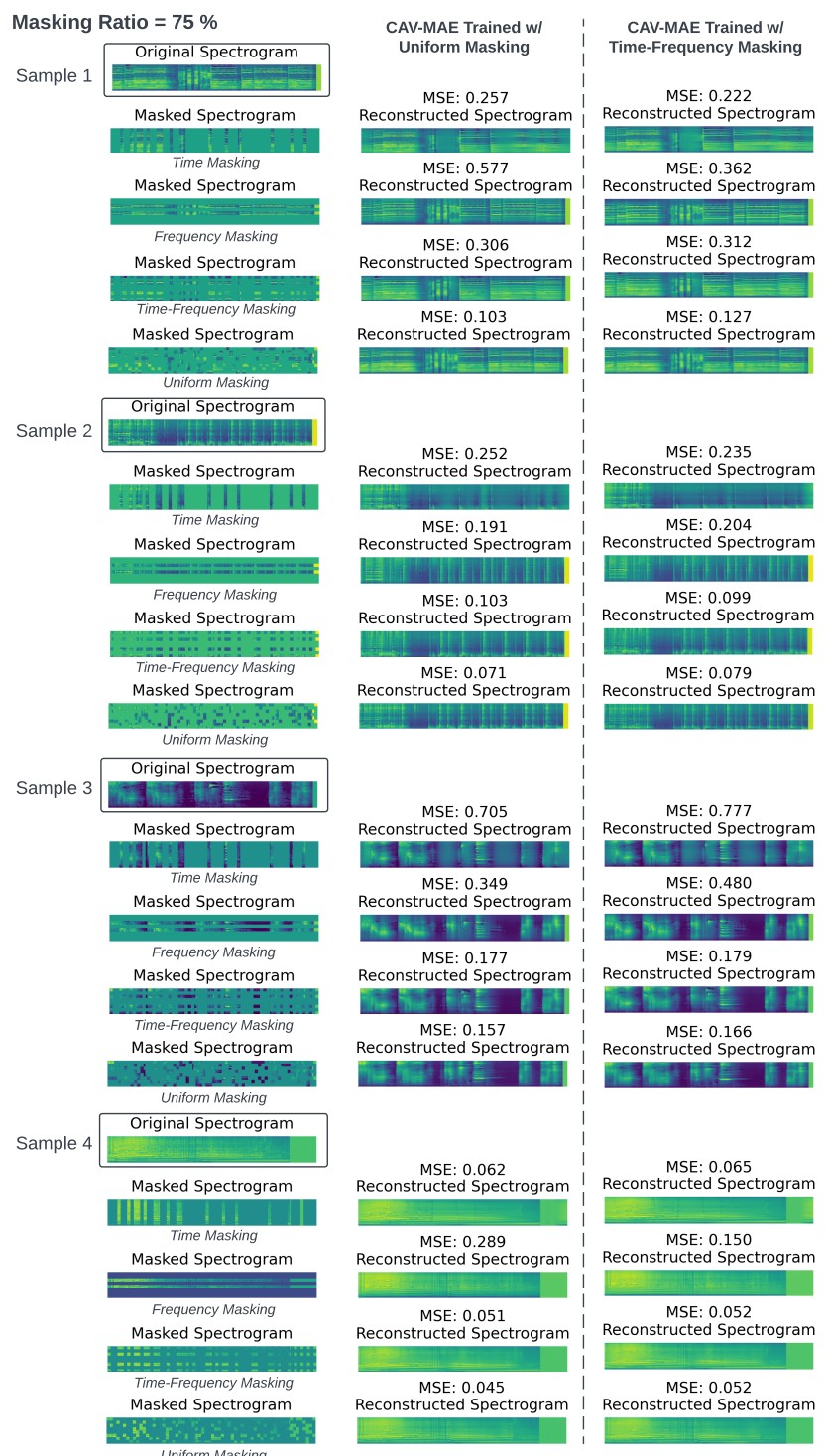

Figure 12: Reconstructed audio spectrograms in various inference masking settings with a 75% masking ratio. We compare the outputs of a CAV-MAE model trained with a uniform, unstructured masking strategy (second column) and a CAV-MAE model trained with a time-frequency masking strategy (third column). Both CAV-MAE models are trained with a 75% masking ratio. Reconstruction mean squared error (MSE) is shown above each reconstructed spectrogram.

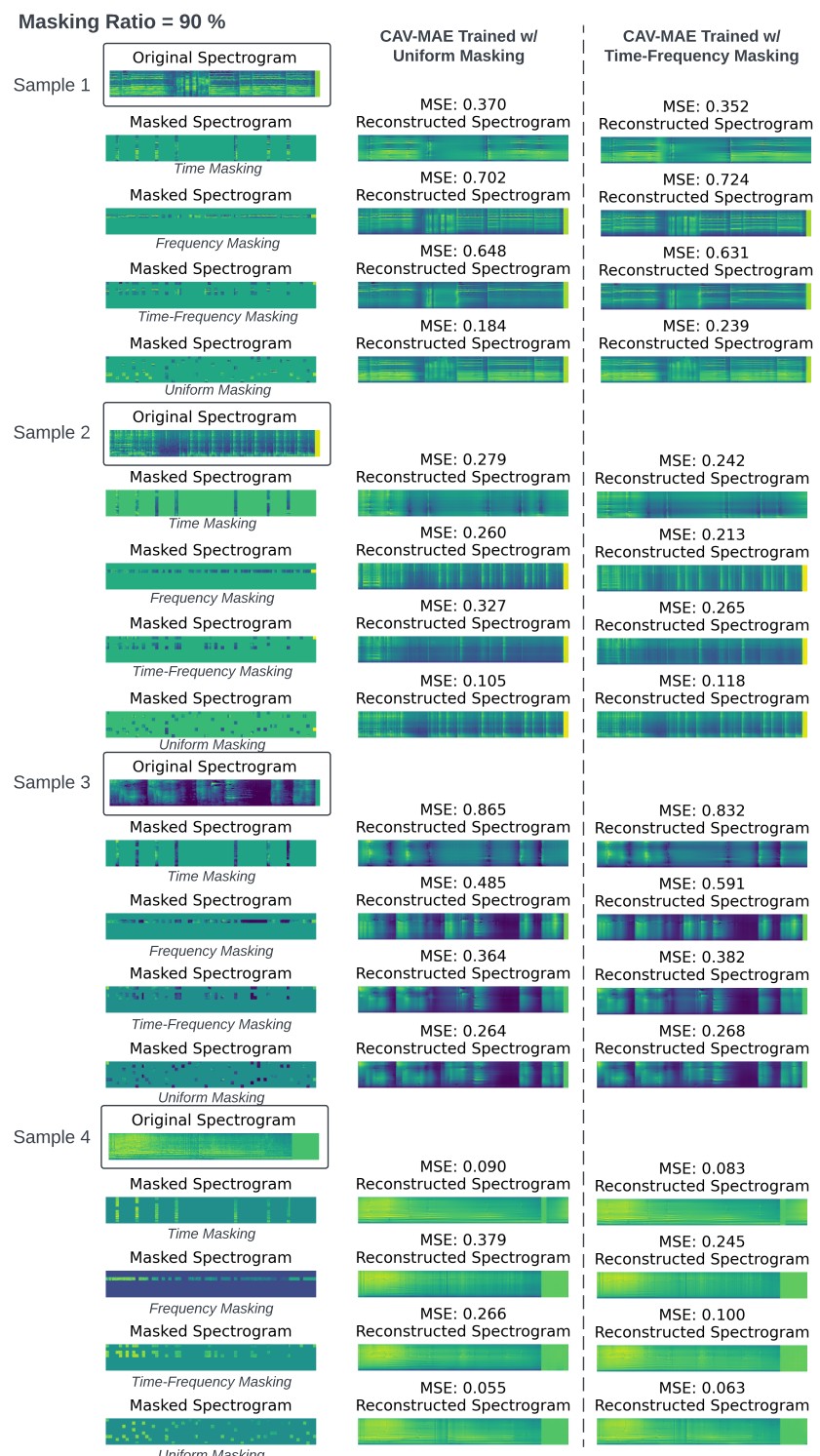

Figure 13: Reconstructed audio spectrograms in various inference masking settings with a 90% masking ratio. We compare the outputs of a CAV-MAE model trained with a uniform, unstructured masking strategy (second column) and a CAV-MAE model trained with a time-frequency masking strategy (third column). Both CAV-MAE models are trained with a 75% masking ratio. Reconstruction mean squared error (MSE) is shown above each reconstructed spectrogram.

