# OpenReview forum: "Contrastive Audio-Visual Masked Autoencoder"
_ICLR.cc/2023/Conference — ICLR 2023 notable top 25%_

### Official Review · Reviewer_oCcH · 2022-10-24

**Confidence:** 5
**Correctness:** 2
**Technical Novelty And Significance:** 2
**Empirical Novelty And Significance:** 3
**Recommendation:** 6

**Clarity, Quality, Novelty And Reproducibility:**

This work is clearly presented, and the results are based on well-known, reproducible baselines including AudioSet and VGG Sound.


**Strength And Weaknesses:**

Strength:
With the advent of MAE and audioMAE, it is natural to extend such framework to multimodal settings.
The motivation for such idea is strong, and the 3 discussion points are valid given the landscape of the current SOTA.

Weakness:
First, the major logic is to show performance improvement brought by CAV-MAE, and empirically verifies it indeed leads to learning better joint representation. However, CAV-MAE underperforming VS. MBT on multimodal AudioSet is a counter example. Maybe as <Li, et al. 2022 AudioTagging>pointed out, the data quality issue of AudioSet would cause a variation, I would suggest the authors look into the granular errors in AudioSet and report the exact train/test numbers to ensure an apple-to-apple fair comparison.

Normalization of different branches are tuned differently, and this looks to me like an engineering effort. I understand normalization is crucial for transformers tuning, but just wondering if there could be a better way rather than setting a fixed number since this severely limits the generalizability of the model.

I get the point of reporting retrieval score on subsets of AudioSet and VGGSound, however this is not a standard dataset for retrieval benchmark, I would suggest the authors reporting on MSR-VTT in a zero shot way.

In appendix and throughout the text, I realized the authors adopted the vanilla MAE masking, which is the random visual-intuitive masking, but as <Huang et al. 2022 AudioMAE> pointed out, this is not necessarily the most suitable way of masking for spectrogram, I would encourage the authors to experiment with various masking strategies.


**Summary Of The Paper:**

This paper extends MAE from single modality to audio and visual modalities by combining contrastive learning and masked prediction pretext tasks. Experiments show the improvements on VGGSound dataset and better retrieval score.


**Summary Of The Review:**

In short, the paper presents a valid idea, but the experiments could be augmented and extended to further strengthen the arguments.

---

> ### Author Response · Authors · 2022-11-20
> **Author Response to Reviewer OCcH (1/3) - Performance Comparison**
>
> Dear Reviewer oCcH,
>
> Thank you for the very insightful and constructive suggestions and questions, please see the following for our point-by-point reply. We also conducted quite many new experiments to address your concerns, which are in the Appendix of the revised paper (please check the updated pdf file).
>
> ---
> **Question 1 - Compare with MBT on AudioSet**
>
> >First, the major logic is to show performance improvement brought by CAV-MAE, and empirically verifies it indeed leads to learning better joint representation. However, CAV-MAE underperforming VS. MBT on multimodal AudioSet is a counter example.
>
> Yes, the main point we want to convey in this paper is that the contrastive objective and masking data modeling objective are complementary - combining both objectives improves the quality of audio-visual joint representation and downstream task performance. We find this conclusion is true for all datasets we experimented with, including the newly added Kinetics-Sounds experiments. Even on the AudioSet-2M that we did not outperform MBT, CAV-MAE (50.5 mAP) still noticeably outperforms CAV (only contrastive objective, 48.1 mAP) and AV-MAE (only MAE objective, 49.6 mAP), see Table 1. Therefore, our main idea is actually supported by the AudioSet experiments.
>
> We totally understand the reviewer’s concern about CAV-MAE not outperforming MBT. This is mainly due to some differences in the training pipeline and settings: First, MBT is trained with much higher computational resources, it takes 8 frames as input simultaneously during training while our model only takes 1 random frame at a time. Since the Transformer has an O(n^2) complexity, MBT’s computation and memory cost are 64 times larger than ours for the visual processing. That’s why MBT’s visual branch performance is stronger than our model. For the audio modality that MBT and our CAV-MAE use the same setting, our CAV-MAE performs better. The positive side of our setting is that our model is trained with 4 GPUs (MBT is trained with 64 GPUs), making it easier to reproduce.
>
> Second, MBT initializes its network using **supervised pretrained** ImageNet weights, while we only initialize our network using **self-supervised pretrained** ImageNet weights. In the revised paper, **Appendix Section E, Table 11**, we quantify the performance difference led by these two initializations - using supervised pretrained ImageNet weights is 3.7 mAP better than using self-supervised pretrained ImageNet weights, which partly explains the reason why MBT performs better on AudioSet.
>
> ---
> **Question 2 - Performance Impact due to Dataset Version**
>
> > Maybe as <Li, et al. 2022 AudioTagging>pointed out, the data quality issue of AudioSet would cause a variation, I would suggest the authors look into the granular errors in AudioSet and report the exact train/test numbers to ensure an apple-to-apple fair comparison.
>
> We thank the reviewer for this very constructive comment. The reviewer is very correct that the dataset version could lead to a performance variation, given the factor that modern models usually have a small performance gap, the performance impact of the dataset version might not be unneglectable. Unfortunately, AudioSet and many other audio-visual datasets are dynamic (YouTube videos) and change with time (videos are removed by the users).
>
> To ​​mitigate this issue, following the reviewer’s comment, we took the following actions:
>
> 1. We have updated **Appendix Section A Dataset Details** to include the exact number of training/evaluation samples for both AudioSet and VGGSound. We also cited <Li, et al. 2022 AudioTagging> in this section.
>
> 2. We will release the class-wise performance and training/test ids we used after the reviewing process in our code repository.
>
> We hope these could help a more fair comparison with other works.

---

> > ### Author Response · Authors · 2022-11-20
> > **Author Response to Reviewer oCcH (2/3) - Normalization & MSR-VTT Experiments**
> >
> > ---
> > **Question 3. Normalization**
> >
> > >Normalization of different branches are tuned differently, and this looks to me like an engineering effort. I understand normalization is crucial for transformers tuning, but just wondering if there could be a better way rather than setting a fixed number since this severely limits the generalizability of the model.
> >
> > We thank the reviewer for the comment. For the joint encoder, we use separated LayerNorms for different modalities (one for audio, one for visual, and one for audio-visual), which is mainly due to the stats of different modality inputs being different. The number of parameters of these LayerNorms is very small compared to the number of parameters of the entire model, and the weights of LayerNorms are completely learned from the data. We found these LayerNorms actually generalize quite well across the datasets, e.g., even in the linear probing setting (when these LayerNorms are frozen), please see Table 10, 6th column from the left, when we train a model on AudioSet and do linear probing on VGGSound, the result is still competitive.
> >
> > We realize that the “Input Norm Mean/STD” in Table 4 might be confusing, these stats are what we used to normalize the audio spectrogram input (so not related to different branches and joint encoder) and calculated based on the entire AudioSet dataset. Audio spectrogram, depending on the generating algorithm, is usually not 0-mean and has a quite large range, these stats are just to make the audio spectrogram 0 mean and 1 std. In addition, we use the same set of normalization stats for all datasets in this paper, and that works fine.
> >
> > ---
> > **Question 4. MSR-VTT Audio-Visual Retrieval**
> >
> > >I get the point of reporting retrieval score on subsets of AudioSet and VGGSound, however this is not a standard dataset for retrieval benchmark, I would suggest the authors reporting on MSR-VTT in a zero shot way.
> >
> > We thank the reviewer for this very constructive suggestion. The reason we used VGGSound for audio-visual retrieval was that it is not in the pretraining dataset (so zero-shot), and the sound source is always visually evident within the video clip in VGGSound (which is done by filtering the videos with a pretrained vision classifier), so it is ideal for the audio-visual retrieval task.
> >
> > Nonetheless, we totally agree with the reviewer that VGGSound is a relatively new dataset and we can benchmark our model on MSR-VTT. In the revised paper, **Appendix Section D.3 MSR-VTT Retrieval Experiments**, we conduct two audio-visual retrieval experiments.
> >
> > 1. We train CAV and CAV-MAE models on the MSR-VTT training set and evaluate them on the MSR-VTT test set. Note that our models are not pretrained on AudioSet. We then compare the retrieval performance with existing works in the same training setting. As shown in **Appendix Table 8**, our CAV-MAE model outperforms existing methods in both directions. In addition, comparing CAV and CAV-MAE, we again find the MAE training objective does not hurt, or even improve the retrieval performance, which is consistent with our findings on other datasets.
> >
> > 2. We conduct a zero-shot retrieval experiment on MSR-VTT following the reviewer’s suggestion. Specifically, we take the AudioSet pretrained models and directly evaluate them on the MSR-VTT test set. The MSR-VTT training set is not used. We then compare our models with existing models. As shown in **Appendix Table 9**, our CAV-MAE model achieves similar results for visual-audio retrieval performance with existing methods but worse for the audio-visual direction. However, existing methods are pretrained with the 100M HowTo100M dataset, while our models are only trained with the 2M AudioSet dataset. With less than 2% of pretraining data, it is quite impressive that our CAV-MAE model achieves similar results for visual-audio retrieval performance with existing methods. Again, CAV-MAE models still have similar or better results compared with CAV models when \lambda_c is the same, demonstrating the MAE and contrastive objective do not conflict.
> >
> > We hope this additional experiment could strengthen our paper.

---

> > > ### Author Response · Authors · 2022-11-20
> > > **Author Response to Reviewer oCcH (3/3) - Impact of Masking Stategy**
> > >
> > > ---
> > > **Question 5 Impact of Masking Strategy**
> > >
> > > >In appendix and throughout the text, I realized the authors adopted the vanilla MAE masking, which is the random visual-intuitive masking, but as <Huang et al. 2022 AudioMAE> pointed out, this is not necessarily the most suitable way of masking for spectrogram, I would encourage the authors to experiment with various masking strategies.
> > >
> > > We thank the reviewer for this very constructive comment. This motivates us to dive deep into the impact of training masking strategies. In the revised paper, we have added substantially more materials regarding the masking strategies. We add a new section **Appendix F Impact of Masking Strategy and Masking Ratio** in the revised paper, where we study two core questions of masking:
> > >
> > > 1. Masking ratio. Throughout the original manuscript, we use a 75% masking ratio for both audio and visual input. This is mainly due to many previous MAE works reporting a masking ratio of ~75% is appropriate for both audio and visual input. However, it is unclear if such a high masking ratio is also appropriate for the contrastive objective. In particular, aggressive augmentation is not commonly used in audio-visual contrastive learning. Therefore, we conduct experiments to check the impact of the training masking ratio for CAV, AV-MAE, and CAV-MAE. As shown in Table 12, we find CAV model does perform slightly well with a smaller masking ratio (50%), but the difference is minor. When the masking ratio is 75%, CAV still performs quite well. This shows high masking ratio works well for both contrastive learning and masked data modeling frameworks, which allows us to unify them into a single CAV-MAE training process.
> > >
> > > 2. Masking strategy. Just as the review points out, throughout the original manuscript, we use a uniform, unstructured masking strategy for both audio and visual input. However, unlike visual modalities, the two dimensions of audio spectrograms are heterogeneous. We thus explore the impact of masking strategies for audio input. Specifically, we apply time, frequency, and time-frequency masking strategies (depicted in Figure 3) and compare them with the uniform unstructured masking strategy. As shown in Table 13, we find that all four training masking strategies lead to similar performance when the training masking ratio is 75%. However, as we show in Figure 5, structured masking strategies make reconstruction more challenging. Therefore, we also pretrain a CAV-MAE model trained with time-frequency masking at a lower masking ratio of 50%, which shows slightly better performance on both AudioSet-20K and VGGSound. However, reducing the training masking ratio from 75% to 50% causes a 4X pretraining computational overhead.
> > >
> > > We also want to clarify that in <Huang et al. 2022 AudioMAE>, masking is used in both pretraining and finetuning. In the pretraining stage, the authors found “unstructured” masking is preferred (figure 4 of <Huang et al. 2022 AudioMAE>), which is why we decided to use unstructured masking in the original manuscript. In the fine-tuning stage, the authors found structured masking is better, but we do not use masking in finetuning for simplicity.
> > >
> > > Finally, in the revised paper, we also studied the impact of the training masking strategy on inference reconstruction ability (**Appendix Section H.2 Audio Spectrogram Reconstruction Under Various Inference Masking Settings, Figure 11, and Figure 12**). In short, the conclusion is that while the training masking strategy only minorly impacts the downstream classification task, it has a relatively large impact on reconstruction.
> > >
> > > We hope this set of new experiments could address the reviewer’s concern and strengthen our paper.
> > >
> > > ---
> > > **Other Improvements**
> > >
> > > Besides **Appendix Section D (MSR-VTT), F (Masking Strategy), and H (Reconstruction under Various Masking)**, we also improve the paper according to other reviewers’ suggestions and comments. Specifically, we test our models on a new dataset Kinetics-Sounds for the audio-visual action recognition task (**Appendix Section C**). CAV-MAE gets good results on it. We also dive deep into model initialization (**Appendix Section E**) and frame rate (**Appendix Section G**). We would be very grateful if the reviewer could also take a read on the new sections and let us know your thoughts (and thoughts on our above responses)!

---

> > > > ### Comment · Reviewer_oCcH · 2022-11-21
> > > > **Thank you so much for the clarification! Raising score!**
> > > >
> > > > Dear authors:
> > > > I am pleasantly surprised by your responses and the additional results you have conducted!
> > > > I think this is the point of rebuttal, which is to encourage us to do better not to put ppl down.
> > > > First and foremost, I am voting **in favor of** the updated paper, **raising my score from 5-> 6**.
> > > > I appreciate the authors' detailed discussion, which I believe is the valuable part:
> > > > **Question 1 - Compare with MBT on AudioSet**:
> > > > I buy your explanation
> > > > **Question 2 - Performance Impact due to Dataset Version**:
> > > > Thank you for the extra results!
> > > > **Question 3. Normalization**
> > > > Though I acknowledge your response, despite the paper's results, my rant (as someone who also works in the field, not affecting my scoring of the paper): I feel like the community still doesn't understand normalization very well, whether batch-norm, layer-norm, or feature-normalization step...think about input matrix, lalyer-norm(row-wise), batch_norm(column-wise), and feature-norm(element-wise), how they store the normalization stats from the trainset and try to generalize to the test set. I wonder if there could be a little more intuition we could draw here, beyond the down-stream empirical score.
> > > > **Question 4. MSR-VTT Audio-Visual Retrieval**
> > > > Thanks for the additional results!

---

> > > > > ### Author Response · Authors · 2022-11-22
> > > > > **Response to Reviewer oCcH - Normalization (Cont.)**
> > > > >
> > > > > Dear Reviewer oCcH,
> > > > >
> > > > > Thank you so much for checking and replying our response promptly and raising the score!
> > > > >
> > > > > Regarding normalization, we have two types of normalization:
> > > > >
> > > > > 1. **Input Normalization:** For images, following the standard processing method, we use mean = [0.485, 0.456, 0.406] and std = [0.229, 0.224, 0.225] for each of the three RGB channels, respectively (torchvision/timm officially use these stats, they are calculated on ImageNet). Similarly, for audios, we calculate the mean (-5.081) and std (4.485) of the spectrogram based on AudioSet, and apply them to all datasets we experiment with. The input normalization is simply x_norm=(x-mean)/std. We find both image and audio norm stats generalize across the datasets, as all results we report in this paper are based on these fixed norm stats, we did not calculate norm stats for each dataset (VGGSound, MSR-VTT, and Kinetics-Sounds).
> > > > >
> > > > > 2. **Intermediate Layer Normalization:** For both audio and visual input, and for all layers, we use Layer Normalization (LN). More specifically, for all Transformer layers, we have two LNs, one before the attention layer, and another before the MLP layer (i.e., Pre-Norm). This is true for the audio encoder, the visual encoder, the joint encoder, and the joint decoder. However, the unique design for the joint encoder is it has three sets of LNs (in total 3*2=6 LNs), one set for audio, one set for visual, and one set for a-v concatenation, the reason is that different modality has different stats. Even in fine-tuning, our linear layers are normalized with LNs. This design makes all intermediate normalization consistent, and this is also the design of `timm` implementation of the vision Transformer and the official implementation of vision MAE. In other words, we do not mix using batch normalization, layer normalization, and feature normalization.
> > > > > On the other hand, all LNs are trainable in pretraining and finetuning. So in the finetuning stage, the LNs can adapt to the (new) downstream dataset - we haven’t quantitatively measured how they change. But we evaluated the linear probing setting, where the LNs are frozen. As shown in Table 10, it seems to be fine and the linear probing results are good when we transfer from AudioSet to VGGSound.
> > > > >
> > > > > We totally agree with the reviewer that such design is mainly empirical / following previous experience, and the optimal normalization strategy worth getting much more attention, particularly in its working mechanism and generalization ability (e.g., which norm transfers best). We thank the reviewer again for the very insightful comment.

---

### Official Review · Reviewer_M9QZ · 2022-10-26

**Confidence:** 4
**Correctness:** 4
**Technical Novelty And Significance:** 2
**Empirical Novelty And Significance:** 3
**Recommendation:** 8

**Clarity, Quality, Novelty And Reproducibility:**

Technically, both MAE and contrastive learning are not new. But the proposed CAV-MAE can nicely unify them in one single framework. The thorough experiments can well validate the proposed approach.

Thorough Implementation details are provided.

**Strength And Weaknesses:**

Pros:

+ The proposed Contrastive Audio-Visual Masked Auto-Encoder (CAV-MAE) can nicely unify MAE and contrastive learning for self-supervised audio-visual learning. In CAV-MAE, unimodal and joint representations are utilized for contrastive and MAE learning.

+ Extensive experimental comparison and ablation studies on AudioSet and VGGSound can validate the effectiveness of the proposed CAV-MAE.

+ I enjoy reading this paper. It is quite easy to follow. The analysis paragraphs can well validate and explain the proposed method.

Cons:

- Not all visual objects make sounds and not all sound sources are visible. The audio-visual mismatch may introduce a lot of false positive audio-visual pairs for audio-visual self-supervised learning. Whether the proposed method can mitigate the data issue? How does the issue affect MAE and contrastive learning respectively? I suggest the authors add discussions on this.

- I was impressed by the reconstructed spectrograms when 75% of the input is masked. Why does audio MAE work? For visual scenes, objects are grouped in certain regions. But spectrograms have quite isolated TF patterns. But, it seems that MAE works for both modalities. Really hope to hear more clarifications and discussions.







**Summary Of The Paper:**

The authors extend the Masked Auto-Encoder (MAE) model from a single modality to audio-visual multi-modalities and combine it with contrastive learning for self-supervised audio-visual learning. Although the two self-supervised learning (SSL) strategies are not new, the proposed Contrastive Audio-Visual Masked Auto-Encoder nicely unifies them in one single framework. Extensive experimental results demonstrate that the two SSL methods are mutually beneficial and the pre-trained models can boost audio-visual classification and cross-modal retrieval performance.

**Summary Of The Review:**

The proposed  CAV-MAE is technically sound and extensive experiments can validate the superiority of the proposed method. I only have a few concerns about the proposed method and I will be happy to upgrade my rating if the authors can address the concerns during the discussion phase.

***Post-rebuttal***

The new experiments and analysis can well address my concerns. I have increased my rating to 8 from 6.

---

> ### Author Response · Authors · 2022-11-19
> **Author Response to Reviewer M9QZ (1/2) False-Positive Pairs & Audio Reconstruction**
>
> Dear Reviewer M9QZ:
>
> Thank you so much for giving us positive feedback and the very insightful questions. Please see our point by point responses below.
>
> ---
> **Question 1 - False-Positive Audio-Visual Pairs in Data**
>
> >Not all visual objects make sounds and not all sound sources are visible. The audio-visual mismatch may introduce a lot of false positive audio-visual pairs for audio-visual self-supervised learning. Whether the proposed method can mitigate the data issue? How does the issue affect MAE and contrastive learning respectively? I suggest the authors add discussions on this.
>
> We thank the reviewer for the very insightful question! Just as the review points out, false positives are a major problem for audio-visual datasets. Theoretically, this would impact contrastive learning more as contrastive learning solely relies on the audio-visual pair as the supervision signal, while AV-MAE also uses the original unmasked inputs as the supervision signal.
>
> In fact, we forgot to mention in the original paper that compared with AudioSet, one advantage of VGGSound is that the sound source is always visually evident within the video clip, which is done by filtering the videos with a pretrained vision classifier (now we added it to **Appendix A Dataset Details**). So we can view VGGSound contains fewer false-positive samples than AudioSet (though for a specific frame, it is totally possible that the sound source is not visually presented, nor it guarantees all visual objects that make sounds). However, it is hard to quantify how many false-positives in each dataset, and disentangling the impact of false-positive pairs from many other factors (e.g., model size, domain difference, etc) that could impact the result is a non-trivial task. We will for sure add a discussion in the paper after we have more mature thoughts and experiments. Thank you again for this comment.
>
> ---
> **Question 2 - Audio Spectrogram Reconstruction**
>
> >I was impressed by the reconstructed spectrograms when 75% of the input is masked. Why does audio MAE work? For visual scenes, objects are grouped in certain regions. But spectrograms have quite isolated TF patterns. But, it seems that MAE works for both modalities. Really hope to hear more clarifications and discussions.
>
> We thank the reviewer for the very insightful question. This motivates us to think deeper about this question. We have added substantially more discussion regarding masking and reconstruction in the revised paper, in **Appendix F and H, Figure 3, 5, 11, 12**. Please see the following:
>
> ---
> *Q1: Why does the model reconstruct the audio spectrogram well with a 75% or even higher masking ratio?*
>
> We believe there are two reasons. First, though a large portion of the audio spectrogram is masked, with the uniform unstructured masking strategy, there are always some unmasked patches around the masked ones, the model can use neighboring unmasked patches to recover masked patches. In order to do so, the model needs to learn the *local* spectrogram structure during training. Second, though audio spectrograms look unorganized, their underlying structure is quite clear, e.g., for human speech, the number of phones is limited. If the model sees a high-frequency component, it is likely to be a consonant (e.g., an “S”) rather than a vowel and thus there are no formants. General audios are much more diverse in acoustic characteristics, but still, follow some rules. In order to do so, the model needs to learn the *global* structure of the spectrogram during training. We hypothesize that the model learns both *local* and *global* spectrogram structure and that's the reason why the model can do well in the reconstruction task.
>
> To verify these points, in the revised paper (please check the new pdf), **Appendix Section H.2 and Figures 3, 11, and 12**. We try to use various masking strategies in inference and observe the reconstruction mean squared error. We find that when we mask a span of time/frequency, the reconstruction task becomes harder for the model (higher MSE), showing that the model indeed requires *local* information for reconstruction. Nonetheless, even in these cases, the output is still quite well, indicating the model has also learned the *global* structure of the spectrogram.
>
> Which masking setting is hardest for the model? In the revised paper, **Figure 5**, we quantify the MSE of the model with various inference masking strategies. On average, time masking is the most difficult, followed by frequency masking, time-frequency masking, and uniform unstructured masking.
>
> (to be continued)

---

> > ### Author Response · Authors · 2022-11-19
> > **Author Response to Reviewer M9QZ (2/2) Audio Reconstruction & Impact of Masking Strategies**
> >
> > (Cont.)
> >
> > ---
> > *Q2: Since we know different masking strategies have different difficulties, can we use masking strategies other than uniform unstructured masking in training? How would that impact the downstream classification task and reconstruction task?*
> >
> > To answer this question, in the new **Appendix F Impact of Masking Strategy and Masking Ratio**, we study the impact of two masking designs:
> >
> > 1. Masking ratio. Throughout the original manuscript, we use a 75% masking ratio for both audio and visual input. This is mainly due to many previous MAE works reporting a masking ratio of ~75% is appropriate for both audio and visual input. However, it is unclear if such a high masking ratio is also appropriate for the contrastive objective. In particular, aggressive augmentation is not commonly used in audio-visual contrastive learning. Therefore, we conduct experiments to check the impact of the training masking ratio for CAV, AV-MAE, and CAV-MAE. As shown in Table 12, we find the CAV model does perform slightly well with a smaller masking ratio (50%), but the difference is minor. When the masking ratio is 75%, CAV still performs quite well. This shows high masking ratio works well for both contrastive learning and masked data modeling frameworks, which allows us to unify them into a single CAV-MAE training process.
> >
> > 2. Masking strategy. Throughout the original manuscript, we use a uniform, unstructured masking strategy for both audio and visual input. However, unlike visual modalities, the two dimensions of audio spectrograms are heterogeneous. In the revised paper, we apply time, frequency, and time-frequency masking strategies (depicted in Figure 3) and compare them with the uniform unstructured masking strategy. As shown in Table 13, we find that all four training masking strategies lead to similar performance when the training masking ratio is 75%. However, as we show in Figure 5, structured masking strategies make reconstruction more challenging. Therefore, we also pretrain a CAV-MAE model trained with time-frequency masking at a lower masking ratio of 50%, which shows slightly better performance on both AudioSet-20K and VGGSound.
> >
> > To summarize, both the masking ratio and strategy do not greatly impact the downstream classification performance. However, we do observe **a large reconstruction performance difference** between a CAV model trained with uniform masking and structured time-frequency masking. As shown in **Figure 5**, the latter almost always does better in audio spectrogram reconstruction except for the uniform masking case. This demonstrates that though the downstream classification performance is similar, different masking strategy actually forces the model to learn differently (learn more *local* spectrogram structure of more *global* spectrogram structure), which reflects in its reconstruction ability. We thank the reviewer again for motivating us on this point.
> >
> > ---
> > **Other Improvements**
> >
> > Besides **Appendix Section F (Masking), H (Reconstruction)** mentioned above, we also improve the paper according to other reviewers’ suggestions and comments. Specifically, we test our model on two new tasks Kinetics-Sounds and MSR-VTT. On both new tasks, our CAV-MAE matches or outperforms SOTA. And we also dive deep into model initialization and frame rate. These are in **Appendix, Section C, D, E, and G**. We hope these make the paper stronger. We would be very grateful if the reviewer could also take a read on the new sections and let us know your thoughts!

---

> > > ### Comment · Reviewer_M9QZ · 2022-11-23
> > > **Thanks for the response!**
> > >
> > > Thank the authors for the response! The new experiments and analysis can well address my concerns. I am happy to increase my score to 8 from 6.

---

> > > > ### Author Response · Authors · 2022-11-23
> > > > **Response to Reviewer M9QZ**
> > > >
> > > > Dear Reviewer M9QZ,
> > > >
> > > > Thank you so much for checking our new experiments and analysis and raising the score. We are glad that they addressed your concerns.

---

### Official Review · Reviewer_qVEJ · 2022-10-30

**Confidence:** 3
**Correctness:** 3
**Technical Novelty And Significance:** 2
**Empirical Novelty And Significance:** 2
**Recommendation:** 6

**Clarity, Quality, Novelty And Reproducibility:**

**Novelty:** As described above (weaknesses), I believe limited novelty is the biggest problem of this paper.

**Clarity/Quality:** The presentation is clear, and the work is easy to be followed.

**Reproducibility**: If the code will be released, I believe this work is easy to reproduce.

**Strength And Weaknesses:**

### Strengths

- The proposed method leverages multi-modal information well with two hot self-supervised ways, contrastive learning, and mask auto-encoder. The method achieves great performance on audio-visual benchmarks.
- The designed method combining contrastive learning and multi-modal MAE is promising for multi-modal tasks.
- Sufficient ablation results are provided.
- This paper is well-written and easy to follow.

### Weaknesses

- This work is not novel enough. It is well-known that contrastive learning and MAE are powerful self-supervised methods, and applying them to a new domain based on relative works is simple yet effective. This paper extends the combination of the two approaches to the multi-modal learning literature, but the architecture is trivial. Besides, this work focuses on the audio-visual task. However, no domain knowledge is leveraged in this work, and the pipeline can be the same as those focusing on text-image multi-modal learning.
- Though lots of experimental results are shown in the paper, the work only focuses on audio-visual event classification and audio-visual retrieval. More relative experiments should be conducted.
- In section 2.3, the authors list some key designs of CAV-MAE. However, these points are all based on the common sense of contrastive learning, MAE, or multi-modal learning. Overall, the framework's design lacks novelty, and the application of the self-supervised methods is so plain. Therefore, the framework's design can be deemed as simple incremental work.

**Summary Of The Paper:**

This paper proposes a multi-modal Masked Auto-Encoder for audio-visual tasks. More specifically, the proposed method combines contrastive learning and masked data modeling for better joint and coordinated audio-visual representation. The model based on self-supervised training achieves a new SOTA on VGGSound and great performance on AudioSet.

**Summary Of The Review:**

This paper combines two powerful self-supervised methods, MAE and contrastive learning, and extends the combination to the audio-visual domain.  The overall pipeline is easy to understand, and the paper is clearly written. However, this paper contributes little to the relative areas, including self-supervised learning and multi-modal learning. And this method is not novelty enough.

---

> ### Author Response · Authors · 2022-11-20
> **Author Response to Reviewer qVEJ (1/2) - Lack of Novelty**
>
> Dear Reviewer qVEJ,
>
> Thank you so much for taking the time to read our paper and providing very valuable comments and suggestions regarding the novelty and experiments of this work. Please see the following for our point-by-point response. Please also note that we have added substantially more materials in the revised paper (please check the updated pdf file).
>
> ---
> **Question 1. Lack of Novelty**
>
> >This work is not novel enough. It is well-known that contrastive learning and MAE are powerful self-supervised methods, and applying them to a new domain based on relative works is simple yet effective. This paper extends the combination of the two approaches to the multi-modal learning literature, but the architecture is trivial. Besides, this work focuses on the audio-visual task.
>
> We understand the reviewer’s concern about the novelty. Just as the reviewer points out, both contrastive learning and masked data modeling have been extensively studied. In addition, text-image multi-modality masked autoencoder has also been previously proposed. Therefore, though our “vanilla” audio-visual masked autoencoder is the first application of MAE in the audio-visual domain, we only list it as a minor contribution.
>
> The main novelty/contribution of this paper is that we are the first to integrate the two major self-supervised frameworks in multi-modal learning and demonstrate such integration helps the model learn strong *joint* (for classification) and *coordinated* (for retrieval) representations at the same time. We conduct extensive experiments to prove this idea. We believe this is a non-trivial finding and want to convey the message to the audio-visual research community that masked data modeling and contrastive learning can be and should be combined together, that’s why we author this paper.
>
> We also want to clarify that while a concurrent work CMAE (Huang et al., 2022) also combines MAE and contrastive loss, it is designed for single-modal images. The implementation of CMAE is very different from our model, it consists of two separated online and target encoders, and two separated feature and pixel decoders. We are not aware of CMAE when we start preparing this work, and the network design does not overlap. In addition, for single modality images, there’s no pair supervision information that can be leveraged. While for multi-modal learning, pair information is a very valuable training signal. Therefore the motivation of the two works is also not identical.
>
> ------
>
> >However, no domain knowledge is leveraged in this work, and the pipeline can be the same as those focusing on text-image multi-modal learning.
>
> The reviewer is correct in that we did not use much (audio and visual) domain knowledge for each modality. Actually, we process the two very different modalities using almost the same method except for input normalization. However, we do not view this as a disadvantage of our model. Using less inductive bias for each modality actually makes the method more generalizable. There are quite many works exploring this direction, e.g., in Perceiver (Jaegle, 2021), the authors use a same architecture for various modalities; in Data2vec (Baevski, 2022), the authors use a same pretraining algorithm for various modalities, etc.
>
> And yes, we agree with the reviewer that our proposed CAV-MAE, with some modification, could be applied to the text-image domain (as well as other multi-modal domains). But to the best of our knowledge, there’s also no model in the text-image domain having a similar design as CAV-MAE that explicitly unifies MAE and contrastive learning in a single training process. Most multi-modal text-image mask autoencoder models are close to our “vanilla” AV-MAE model. Thus, we do not directly adopt a text-image model to the audio-visual domain, but design a unique CAV-MAE model and apply it to the audio-visual domain in this paper, which could be potentially used in other multi-modal applications in the future.

---

> > ### Author Response · Authors · 2022-11-20
> > **Author Response to Reviewer qVEJ (2/2) New Experiments & Network Design**
> >
> > **Question 2. More Experiments**
> >
> > ------
> > >Though lots of experimental results are shown in the paper, the work only focuses on audio-visual event classification and audio-visual retrieval. More relative experiments should be conducted.
> >
> > We thank the reviewer for the very constructive suggestion. In the revised paper (please kindly check the updated pdf file), we have added substantially more experiments, including:
> >
> > 1. Experiments on audio-visual action recognition task (Kinetics-Sounds dataset). Our CAV-MAE model matches the SOTA MBT model. (**Appendix C Audio-Visual Action Recognition Experiments**)
> > 2. Experiments on MSR-VTT dataset for the audio-visual bi-directional retrieval task. Our CAV-MAE model outperforms the SOTA models. (**Appendix D.3 MSR-VTT Dataset Retrieval Experiments**)
> > 3. Experiments on the performance impact of model initialization strategies. (**Appendix E Impact of Model Initialization**)
> > 4. Experiments on the performance impact of training masking strategies. (**Appendix F Impact of Masking Strategy and Masking Ratio**)
> > 5. Experiments on the performance impact of frame rate. (**Appendix G Performance Impact of the Number of Frames Used**)
> > 6. Experiment on the reconstruction ability of models trained with various masking strategies. (**Appendix H.2 Audio Spectrogram Reconstruction Under Various Inference Masking Settings**)
> >
> > These new experiments either test our model on new tasks/datasets, or dive deep into the key components of the model design. We hope they can partly address the reviewer’s concern about the experiments.
> >
> > ------
> > **Question 3. Network Design**
> >
> > >In section 2.3, the authors list some key designs of CAV-MAE. However, these points are all based on the common sense of contrastive learning, MAE, or multi-modal learning. Overall, the framework's design lacks novelty, and the application of the self-supervised methods is so plain. Therefore, the framework's design can be deemed as simple incremental work.
> >
> > We thank the reviewer for the comment. We agree with the reviewer that these designs are based on the common sense of contrastive learning and masked data modeling, but integrating them together requires some unique design (e.g., multiple-stream to avoid contrastive loss collapsing) and trial and error. We did not add unnecessary complex components to the network and viewed its simplicity and straightforwardness as advantages. But to the best of our knowledge, the CAV-MAE architecture is unique, we did not see a very similar model in the audio-visual domain and image-text domain that explicitly unifies MAE and contrastive learning in a single training process.
> >
> > ---
> >
> > We thank the reviewer again for the very valuable comments and would be very grateful if the reviewer could also take a read on the new Appendix Sections (C-H) in the revised paper and let us know your thoughts (and thoughts on our above responses)!

---

> > > ### Comment · Reviewer_qVEJ · 2022-11-23
> > > **Post-Rebuttal**
> > >
> > > Dear authors:
> > >
> > > Thank you for the detailed reply! I've carefully gone through the revised part in the main text and supplementary material, which makes this work stronger. And I thank the authors for addressing my concerns, especially the major issue about the novelty. I misunderstand that the main novelty lies in adapting MAE itself. I strongly recommend the authors to make this part clearer and supplement the additional experiments to the revised version. Based on this, I've raised my score from 3 to 6.
> > >
> > > Best

---

> > > > ### Author Response · Authors · 2022-11-23
> > > > **Response to Reviewer qVEJ**
> > > >
> > > > Dear Reviewer qVEJ,
> > > >
> > > > Thank you so much for carefully checking our responses and raising the score!
> > > >
> > > > >I strongly recommend the authors to make this part clearer and supplement the additional experiments to the revised version.
> > > >
> > > > Yes, the additional experiments are already in the revised pdf manuscript (Appendix). We want to thank you again for pointing out the clarity of the novelty issue, which pushes us to make a more clear discussion, and for sure we will incorporate the discussion in the above responses in the *Introduction* and *Related Work* section. We hope that would more precisely highlight the contribution of this work and better place it in the context.

---

### Official Review · Reviewer_TNd7 · 2022-10-31

**Confidence:** 5
**Correctness:** 3
**Technical Novelty And Significance:** 2
**Empirical Novelty And Significance:** 3
**Recommendation:** 6

**Clarity, Quality, Novelty And Reproducibility:**

The paper is clear and well-written. The novelty might appear limited as it combines two well known methods in straightforward manner, I believe this is not a major concern here and the approach is neat.

**Strength And Weaknesses:**

Strength
1. The proposed approach is simple and logical. It tried to combine strengths of masked data modeling and contrastive learning and applied it to audio-visual settings where they have not been explored.


2. The paper is very neatly written and easy to read and follow.


3. The results on the datasets used show good performance improvements.



Weakness


1. A more exhaustive set of audio-visual tasks would have been better.


2. Some other weaknesses are pointed out in the review below.


**Summary Of The Paper:**

This paper combines  masked auto-encoder based learning and contrastive learning for self-supervised learning in audio-visual settings. The method referred to as Contrastive Audio-Visual Masked Auto-Encoder learning is used to pre-train the models on audiovisual Audioset dataset and then fine-tuned on Audioset and VGGSound. Performance improvements in the downstream tasks of event classification are shown.

**Summary Of The Review:**

The paper combines masked auto-encoder based learning with contrastive learning. It develops and investigates the method for audio-visual representation learning.


1. I believe initializing the weights from MAE models leaves a few things unclear. Performance of the model by training from scratch should be reported. Its unclear how critical such initialization is. Especially, in the current context with MAE based initialization - it becomes a two stage training process. First train each modality with MAE and then combine them using the current approach.


2. The downstream tasks are limited and other audio-visual tasks should be included. CUrrently the paper just focuses on event classification on two datasets. These two datasets do not provide a comprehensive understanding of audio-visual representation learning. Diverse set of tasks like audio-visual action classification, Epic Kitchen benchmarks, audio-visual segmentation/localization etc. would be able to provide a clearer picture of how good the SSL approach is. In fact, Audioset is not the best dataset for audio-visual learning of sounds. A large portion of visual data in Audioset is just redundant (black frames, audio overlayed over random non-relevant videos and so on). Moreover, Audioset is used for pre-training as well. So finetuning results especially on the full Audioset is not an ideal set of experiments.



3. Since the tasks are primarily audio tasks, references to some missing audio and speech SSL works are missing (like Wav2Vec and Hubert). [R1] in particular used wav2vec like learning for non-speech audio representation learning and has results on Audioset and several other audio tasks.



4. In Section 4.3, authors comment on computational efficiency of the proposed method. I am not sure those arguments are convincing. The models are exactly the same and the computational expense of these models will be the same.


R1: Conformer-Based Self-Supervised Learning for Non-Speech Audio Tasks in IEEE ICASSP 2022.

---

> ### Author Response · Authors · 2022-11-19
> **Author Response to Reviewer TNd7 (1/3) - Modal Initialization**
>
> Dear Reviewer TNd7,
>
> Thank you for the positive feedback and all these very valuable and constructive questions/suggestions! Let us respond to your questions point by point.
>
> ---
> **Question 1 - Model Initialization**
>
> >I believe initializing the weights from MAE models leaves a few things unclear. Performance of the model by training from scratch should be reported. Its unclear how critical such initialization is.
>
> Yes, we should report the results without initialization. In this revision (please see the new pdf paper, it has been updated), we add a new section **“Appendix E Impact of Model Initialization”**, which dives deep into the model initialization and pretraining. More specifically, in Table 10, we tested all model initialization / in-domain pretraining combinations, and report both end-to-end finetuning and linear probing results on the three datasets (due to time/computing limit, for AudioSet-2M, we only tested two combinations).
>
> The conclusion is, ImageNet initialization consistently leads to a performance improvement, no matter in the fine-tuning or linear probing setting. However, such improvement saturates with a larger in-domain pretraining dataset, e.g., without ImageNet initialization, CAV-MAE performs just 1.0 mAP lower on AudioSet-2M. Therefore, ImageNet initialization is not an indispensable component of the proposed CAV-MAE pretraining framework. Also, as expected, AudioSet CAV-MAE pretraining (our main proposed method) leads to a much larger improvement over the from-scratch model than ImageNet initialization (comparing the second and third row with the first row in Table 10).
>
> The main reason we use ImageNet initialization is we need to get the best results to compare with previous efforts, which typically initialize their network with (supervised) ImageNet pretrained weights (e.g., MBT). As we mentioned in the paper, we use self-supervised pretrained weights to avoid using any ImageNet labels and make our pipeline fully self-supervised. As a cost, self-supervised pretrained weights usually are not as strong as supervised pretrained weights. In this revision, we add Table 11, which quantifies the downstream performance gap between using supervised and self-supervised pretrained weights. On AudioSet-20k, initializing the model with supervised pretrained weights leads to a 3.7 mAP improvement over self-supervised pretrained weights. We hope this new result could help the reader better understand the difference between these two settings. This could also partly explain the small performance gap between our CAV-MAE and the SOTA MBT model.
>
> ---
> >Especially, in the current context with MAE-based initialization - it becomes a two-stage training process. First, train each modality with MAE and then combine them using the current approach.
>
> This is very correct, and we desire not to have a two-stage process as we already unified contrastive learning and masked data modeling in a single architecture and training process. But as we mentioned above, the main reason for this initialization setting is to fairly compare with existing models. On the other hand, we directly get the weight from the original vision MAE model from their public release, so practically, we didn’t pretrain the ImageNet model by ourselves and only have a single-stage CAV-MAE pretraining process, but conceptually, the reviewer is correct (please check Appendix Section E for more details).

---

> > ### Author Response · Authors · 2022-11-19
> > **Author Response to Reviewer TNd7 (2/3) - More Comprehensive Experiments**
> >
> > ---
> > **Question 2 - More Comprehensive Experiments**
> >
> > >The downstream tasks are limited and other audio-visual tasks should be included. Currently the paper just focuses on event classification on two datasets. These two datasets do not provide a comprehensive understanding of audio-visual representation learning. Diverse set of tasks like audio-visual action classification, Epic Kitchen benchmarks, audio-visual segmentation/localization etc. would be able to provide a clearer picture of how good the SSL approach is. In fact, Audioset is not the best dataset for audio-visual learning of sounds. A large portion of visual data in Audioset is just redundant (black frames, audio overlayed over random non-relevant videos and so on). Moreover, Audioset is used for pre-training as well. So finetuning results especially on the full Audioset is not an ideal set of experiments.
> >
> > We thank the reviewer for the very constructive comments! We agree with the reviewer that audio-visual event classification and audio-visual bi-directional retrieval are just part of the large audio-visual research field. The motivation for us to include these two tasks is that we want to have 1) a joint audio-visual classification task to demonstrate CAV-MAE is good at learning an audio-visual *joint* representation, and 2) a retrieval task to demonstrate CAV-MAE also learns a good audio-visual *coordinated* representation at the same time.
> >
> > For audio-visual joint representation learning, the reason for us to choose AudioSet and VGGSound is multifold:
> > 1. AudioSet and VGGSound are relatively “modality-balanced”, i.e., the audio and visual branches perform relatively similarly, and using two modalities is obviously better than just using one. Therefore, they are ideal for evaluating the audio-visual fusion ability. In contrast, some other audio-visual datasets are dominated by one modality (i.e., the performance of one branch is much better than another, and fusing two modalities leads to minor improvement).
> > 2. The SOTA audio-visual model MBT is mainly benchmarked on these two datasets.
> > 3. Previous audio MAE works are mostly benchmarked on AudioSet, and we want to compare with them to demonstrate CAV-MAE multi-modal pretraining improves audio single-modal performance. Further, the SOTA Audio-MAE (Xu et.al. ,2022) also conducts both pretraining and finetuning on AudioSet-2M.
> >
> > Nonetheless, we totally agree with the reviewer that adding more tasks would make our results more solid. Among the datasets/tasks the reviewer suggests, we found Kinetics-Sounds is an appropriate dataset because its action classes are selected to be potentially manifested visually and aurally. In contrast, the Kinetics-400/Epic-Kitchen datasets are more dominated by the visual modality. We conduct experiments on Kinetics-Sounds and add a new section **“Appendix C Audio-Visual Action Recognition Experiments”**. In general, our CAV-MAE model matches the SOTA MBT model on Kinetics-Sounds, and CAV-MAE also performs better than CAV and MAE on Kinetics-Sounds. Besides Kinetics-Sounds, we also add another retrieval task on MSR-VTT (**Appendix D.3 MSR-VTT Dataset Retrieval Experiments**), on MSR-VTT, our CAV-MAE outperforms the SOTA models. We hope these new experiments will make our paper stronger.
> >
> > The reviewer also suggests the audio-visual localization task, which is a piece of very practical and nice suggestion. Considering the VGGSound-SS dataset (a subset of VGGSound) does contain bounding box annotation, we should be able to test this task. In addition, it is another task besides retrieval that can show the quality of audio-visual coordinated representation. Nonetheless, the audio-visual localization pipeline is very different from the one we use, and applying a Transformer-based model for this task is non-trivial (currently, most localization models are CNN based). With the limited time we have, we would consider leaving it as future work.

---

> > > ### Author Response · Authors · 2022-11-19
> > > **Author Response to Reviewer TNd7 (3/3) - Computational Resources and Minors**
> > >
> > > ---
> > > **Question 3 - Missing References**
> > >
> > > >Since the tasks are primarily audio tasks, references to some missing audio and speech SSL works are missing (like Wav2Vec and Hubert). [R1] in particular used wav2vec like learning for non-speech audio representation learning and has results on Audioset and several other audio tasks.
> > > R1: Conformer-Based Self-Supervised Learning for Non-Speech Audio Tasks in IEEE ICASSP 2022.
> > >
> > > We thank the reviewer for pointing the missing references out. We have added wav2vec2.0 HuBERT, and [R1] to the **Related Work** section.
> > >
> > > ---
> > > **Question 4 - Computational Resources**
> > >
> > > >In Section 4.3, authors comment on computational efficiency of the proposed method. I am not sure those arguments are convincing. The models are exactly the same and the computational expense of these models will be the same.
> > >
> > > We thank the reviewer for pointing this out. The reviewer is very correct that the computational overhead of the model is the same as base MAE models (of the same size) in the same setting (e.g., input sequence length, number of epoches). We have clarified in the revised paper, “**Appendix B Training Details**” that our model is not smaller, nor faster than a base MAE model or previous Transformer models (e.g., MBT). We hope this could avoid the potential misunderstanding.
> > >
> > > However, what we meant was, for the same Transformer-based model architecture, training with larger batch size and more epochs usually leads to better performance. This is particularly important for a contrastive learning model because we cannot use gradient accumulation to simulate large batch sizes. One piece of evidence is that the CAV-MAE model trained with 108 batch size for 25 epochs is noticeably better than the CAV-MAE trained with 48 batch size for 12 epochs (Table 1, bottom two rows) while they have exactly the same architecture and size.
> > >
> > > On the other hand, for the same Transformer model, the input sequence length could dramatically change the computational cost. E.g., inputting multiple RGB frames to the Transformer could improve the performance, but at a cost of O(n^2) complexity increase. MBT uses 8 RGB frames as input (input sequence length of 1568), while we only use one at a time (input sequence length of 196), therefore, MBT's visual processing is 8^2=64 times more expensive than ours.
> > >
> > > Most SOTA models are trained with industry-level resources (e.g., 32 TPUs for Perceiver (Jaegle et al., 2021), 64 GPUs for Audio-MAE (Xu et al., 2022) and MBT (Nagrani et al., 2021)), which brings many benefits such as large batch size (particularly useful for contrastive learning), multiple frames input (MBT uses 8 frames as input), and more training epochs (Audio-MAE pretrain the model for 32 epochs). But the positive side of using only 4 GPUs in this work is that the model can be reproduced (both training and inference) by researchers without access to industry-level computing. As promised in the paper, we will release the code and pretrained model after the review process.
> > >
> > > ---
> > > **Other Improvements in the Revision**
> > >
> > > Besides Appendix Section C (**Kinetics-Sounds**), D (**MSR-VTT Retrieval**), and E (**Model Initialization**) mentioned above, we also improve the paper according to other reviewers’ suggestions and comments. Specifically, we also dive deep into masking ratio/strategy, frame rate, and audio spectrogram reconstruction. These are in **Appendix, Section F, G, H**. We would be very grateful if the reviewer could also take a read on the new sections and let us know your thoughts (and thoughts on our above responses)!

---

> > > > ### Author Response · Authors · 2022-12-09
> > > > **Author Response to Reviewer TNd7 - Updated Results for Table 10**
> > > >
> > > > Dear Reviewer TNd7,
> > > >
> > > > After the November 18th paper updating deadline, we are able to finish the remaining two pretraining setting combinations on the full AudioSet-2M in Table 10. Please see the following table for the updated results (new results highlighted in _italics_).
> > > >
> > > > | ImageNet Initialization | AudioSet Pretraining | AudioSet-20K |  AudioSet-20K  | VGGSound (200K) | VGGSound (200K) | AudioSet-2M |
> > > > |:-----------------------:|:--------------------:|:------------:|:--------------:|:---------------:|:---------------:|:-----------:|
> > > > |                         |                      |  Fine Tuning | Linear Probing |   Fine Tuning   |  Linear Probing | Fine Tuning |
> > > > |            No           |          No          |      8.0     |       2.4      |       42.4      |       10.3      |    _33.5_   |
> > > > |           SSL           |          No          |     25.6     |      10.3      |       62.1      |       34.3      |    _47.3_   |
> > > > |            No           |          SSL         |     37.3     |      29.1      |       62.7      |       53.0      |     49.5    |
> > > > |           SSL           |          SSL         |   **40.6**   |    **29.8**    |     **65.4**    |     **54.2**    |   **50.5**  |
> > > >
> > > > The conclusion is consistent for all datasets, including the large AudioSet-2M:
> > > > 1. ImageNet initialization consistently leads to a performance improvement, no matter in the fine-tuning or linear probing setting. However, such improvement saturates with a larger in-domain pretraining dataset, e.g., without ImageNet initialization, CAV-MAE performs just 1.0 mAP lower on AudioSet-2M. Therefore, **ImageNet initialization is not an indispensable component of the proposed CAV-MAE pretraining framework.**
> > > > 2. AudioSet CAV-MAE pretraining (our main proposed method) leads to a much larger improvement than ImageNet initialization, particularly in the linear probing setting and for small datasets  (comparing the second and third row with the first row in the Table).
> > > >
> > > > The main reason we use ImageNet initialization is that we need to get the best results to compare with previous efforts, which typically initialize their network with (supervised) ImageNet pretrained weights (e.g., MBT). As we mentioned before, we use self-supervised pretrained weights to avoid using any ImageNet labels and make our pipeline fully self-supervised. As a cost, self-supervised pretrained weights usually are not as strong as supervised pretrained weights. In Table 11, we quantified the downstream performance gap between using supervised and self-supervised pretrained weights. On AudioSet-20k, initializing the model with supervised pretrained weights leads to a 3.7 mAP improvement over self-supervised pretrained weights.
> > > >
> > > > We thank the reviewer again for raising the ImageNet initialization question, which pushes us to clarify this important point. If you have any further questions or comments, please let us know, we are glad to check and respond to them.

---

### Official Review · Reviewer_Ee8Z · 2022-11-03

**Confidence:** 3
**Correctness:** 3
**Technical Novelty And Significance:** 4
**Empirical Novelty And Significance:** 4
**Recommendation:** 8

**Clarity, Quality, Novelty And Reproducibility:**

The paper is clear enough. The idea of using masked-autoencoding for audio and video is novel. Combining both losses is interesting. The results woud be reproducable if the model is made available.

**Details Of Ethics Concerns:**

No ethics concerns.

**Strength And Weaknesses:**

Strengths:

1. The model seems to be a novel audio-visual self-supervisedly trained autoencoding network that uses both masked autoencoder and contrastive losses. Previous work I believe only used contrastive/coincidence losses for audio-visual learning. Masked autoencoding was only used for each modality separately.
2. The joint encoder is run separately for each modality for the contrastive loss, but it is run jointly for the masked-prediction loss which makes it possible to use for each modality separately too.

Weaknesses:
1. The video is only represented with 10 frames from 10 seconds (1 Hz) which seems low. So, the model probably relies on images rather than the motion.


**Summary Of The Paper:**

A contrastive audio-visual masked autoencoder model is introduced in the paper. The training of self-supervised model is done with both contrastive and masked-autoencoder losses.

The model architecture is transformer and only unmasked inputs are kept as input for the encoder. There is a separate encoder for each modality as well as a joint encoder. The joint encoder can be run for one or both modalities (with its corresponding layer-norms). In the case when the model is fine-tuned for one of the modalities, the joint decoder would only be run with only that modality input.

**Summary Of The Review:**

The paper is an interesting read which generalized masked-autoencoders to multiple modalities of audio and video. In addition a contrastive loss is also used.

Relevant ablations are done by comparing with a vanilla VAE which does not have a contrastive loss. The models are used in classification and retrieval problems and the benefits for CAV-MAE model is shown.

---

> ### Author Response · Authors · 2022-11-19
> **Author Response to Reviewer Ee8Z**
>
> Dear Reviewer Ee8Z:
>
> Thank you for taking the time to consider our paper and giving us positive feedback!
>
> ---
> Regarding your question,
>
> >The video is only represented with 10 frames from 10 seconds (1 Hz) which seems low. So, the model probably relies on images rather than the motion.
>
> Yes, it is a very good question. First, one reason for us to use a relatively low FPS (10 frames per 10s video clip, 1FPS) is that we want to make a fair comparison with prior works, e.g., the MBT model also uses 1FPS. We want to demonstrate that the good performance is from the new model architecture and training objective, rather than a higher FPS. Second, we do find using more frames leads to a performance improvement, as shown in Table 2(h), using multiple (10) frames leads to a 1.1 mAP improvement over using the single middle frame. To respond to your question, in this revision, we further add a section **Appendix H and Figure 4** to check the relationship between performance and FPS in more detail. On the Kinetics-Sounds dataset, we have tested a higher FPS of up to 3 FPS (32 frames for 10s video clips).  We find the performance consistently improves with more frames being used, but the improvement saturates with the increase of frames.
>
> In addition, we also tested our model on a new audio-visual action recognition task - Kinetics-Sounds, which targets recognizing human actions (so more dynamic than scenes). Our CAV-MAE model also matches the SOTA model on this task. Please see **Appendix Section C Audio-Visual Action Recognition Experiments**.
>
> Having said that, we agree with the reviewer that our model relies on images more than motions because we do not have an explicit mechanism to model the change between the frames. That’s why we do not claim our model is able to capture motion. Actually, we focus more on the interaction between the two modalities - audio and visual, and we show that our new self-supervised learning objective can learn a joint and coordinated audio-visual representation. As a first step, we focus more on audio-image models, but CAV-MAE certainly has the potential to be extended to audio-motion modeling by replacing the visual branch with a more motion-specific one. Since the current paper is already quite long, we would hope to leave this as future work.
>
> ---
> **Other Improvements**
>
> Besides **Appendix Section C (action recognition), G, and Figure 4 (frame rate)**, we also improve the paper according to other reviewers’ suggestions and comments. Specifically, we test retrieval on a new dataset MSR-VTT, CAV-MAE achieves SOTA on it. We also dive deep into model initialization, masking ratio/strategy, and audio spectrogram reconstruction. These are in **Appendix, Sections D, E, F, G, H**. We would be very grateful if the reviewer could also read the new sections and let us know your thoughts (and thoughts on our response regarding the motion question)!

---

> > ### Comment · Reviewer_Ee8Z · 2022-12-02
> > **Post-rebuttal note from reviewer**
> >
> > Thanks for your response. The paper have improved. I keep my original good rating.

---

> > > ### Author Response · Authors · 2022-12-09
> > > **Response to Reviewer Ee8Z**
> > >
> > > Dear Reviewer Ee8Z,
> > >
> > > Thank you so much for checking our response and the revised paper. We are glad that you keep the original positive rating.

---

### Author Response · Authors · 2022-11-20
**Author Response & Revision**

Dear Reviewers,

We appreciate your immensely helpful, insightful, and constructive comments on our paper
*Contrastive Audio-Visual Masked Autoencoder*. Following the reviewers' suggestions, we have made a major revision of the paper and supplemented a set of new experiments to address the reviewers' concerns. Please kindly check the **updated pdf file**. To make it easier to track the new sections, we temporarily put all of them in Appendix but will consider moving them to the main manuscript in the next version.

---
1. Following Reviewer TNd7 and qVEJ’s suggestions, we added experiments on the audio-visual action recognition task (Kinetics-Sounds dataset). Our CAV-MAE model matches the SOTA MBT model. (**Appendix C Audio-Visual Action Recognition Experiments**)
2. Following Reviewer oCcH and qVEJ’s suggestions, we added experiments on the MSR-VTT dataset for the audio-visual bi-directional retrieval task. Our CAV-MAE model outperforms the SOTA models. (**Appendix D.3 MSR-VTT Dataset Retrieval Experiments**)
3. Following Reviewer TNd7’s suggestion, we added experiments on the performance impact of model initialization strategies. (**Appendix E Impact of Model Initialization**)
4. Following Reviewer oCcH and M9QZ’s suggestions, we added experiments on the performance impact of training masking strategies. (**Appendix F Impact of Masking Strategy and Masking Ratio**)
5. Following Reviewer Ee8Z’s suggestions, we added experiments on the performance impact of frame rate. (**Appendix G Performance Impact of the Number of Frames Used**)
6. Following Reviewer M9QZ’s suggestions, we added experiments on the reconstruction ability of models trained with various masking strategies. (**Appendix H.2 Audio Spectrogram Reconstruction Under Various Inference Masking Settings**)
---
We hope the revised manuscript will better suit the ICLR conference, and are happy to discuss further with the reviewers and consider further revisions. We thank you for your continued interest in our research. In the following, under each reviewer's comment, we address the concerns of the reviewers point by point.

---

### Decision · Program_Chairs · 2023-01-20

**Decision:**

Accept: notable-top-25%

**Justification For Why Not Higher Score:**

This is a very interesting yet not ground-breaking work.

**Justification For Why Not Lower Score:**

The proposed method extends the well-known masked auto-encoder method for processing audio-visual data and combines contrastive loss with masked data modelling. The experimental results are strong.

**Metareview: Summary, Strengths And Weaknesses:**

The main contributions of the paper are twofold 1) to extend the recently-proposed, well-known masked auto-encoder (MAE) for audio-visual multi-modalities and 2) to combine contrastive learning with masked prediction objective to learn a joint and coordinated representation.  Extensive experiments validate the effectiveness of the proposed method named contrastive audio-visual masked auto-encoder (CAV-MAE), providing new SOTA performance on selected benchmark datasets. Various ablation studies have also performed to help understand the characteristics of the proposed method.  The paper is well-written and easy to follow.

The paper received unanimous acceptance from 5 confident reviewers with scores 86686. The authors were very responsive in answering the reviewers' comments and in conducting extensive extra experiments as requested. As a result, 3 out of 5 reviewers raised their scores during the rebuttal process.

I am recommending acceptance.


**Note From Pc:**

if the above contains the word "oral" or "spotlight" please see: "oral" presentation means -> notable-top-5% and "spotlight" means -> notable-top-25%. As stated in our emails, we are disassociating presentation type from AC recommendations